# Convergent evolution of viral-like Borg archaeal extrachromosomal elements and giant eukaryotic viruses

Jillian F. Banfield [1,2,3,4] ✉, Luis E. Valentin-Alvarado[1,2,5], Ling-Dong Shi[2], Colin Michael Robinson [5], Rebecca S. Bamert[1], Fasseli Coulibaly [2], Zachary K. Barth[6], Frank O. Aylward[6], Marie C. Schoelmerich [2,7], Shufei Lei [2], Rohan Sachdeva [2] & Gavin J. Knott [1] ✉

Borgs are huge extrachromosomal elements of anaerobic methane-oxidizing archaea. They exist in exceedingly complex microbiomes, lack cultivated hosts and have few protein functional annotations, precluding their classification as plasmids, viruses or other. Here, we use in silico structure prediction methods to investigate potential roles for ~10,000 Borg proteins. Prioritizing analysis of multicopy genes that could signal importance for Borg lifestyles, we uncover highly represented de-ubiquitination-like Zn-metalloproteases that may counter host targeting of Borg proteins for proteolysis. Also prevalent are clusters of multicopy genes for production of diverse glycoconjugates that could contribute to decoration of the host cell surface, or of putative capsid proteins that we predict multimerize into pentagonal and hexagonal arrays. Features including megabase-scale linear genomes with inverted terminal repeats, genomic repertoires for energy metabolism, central carbon compound transformations and translation, and pervasive direct repeat regions are shared with giant viruses of eukaryotes, although analyses suggest that these parallels arose via convergent evolution. If Borgs are giant archaeal viruses they would fill the gap in the tri(um)virate of giant viruses of all three domains of life.

Assigning function to proteins of extrachromosomal elements (ECEs) of Archaea, the third Domain of life, is challenging[1]. This is likely due to historical bias towards the characterization of proteins from model organisms that do not represent the staggering sequence diversity acquired over billions of years of evolution. Lack of understanding about archaeal ECEs is important, given that Archaea play key roles in the methane, nitrogen and carbon cycles. Further, as Archaea are the likely ancestors to Eukaryotes[2],

their ECEs may illuminate the evolutionary origin of eukaryotic viruses. Of particular interest to us are Borgs, huge ECEs that replicate in anaerobic methane-oxidizing archaea[3]. Borgs and their host *Methanoperedens* archaea each account for only a miniscule fraction of the DNA in the soil in which they occur. *Methanoperedens* have not yet been obtained in pure culture and all laboratory *Methanoperedens* enrichments lack Borgs. Thus, all information about Borgs must be acquired from their nucleic acid

[1]Biomedicine Discovery Institute, Monash University, Clayton, VIC, Australia. [2]Innovative Genomics Institute, UC Berkeley, Berkeley, California, USA. [3]Earth and Planetary Science, UC Berkeley, Berkeley, California, USA. [4]Environmental Science, Policy and Management, UC Berkeley, Berkeley, California, USA. [5]Plant and Microbial Biology, UC Berkeley, Berkeley, California, USA. [6]Virginia Polytechnic Institute and State University, Blacksburg, VA, USA. [7]Present address: Department of Environmental Systems Sciences, ETH Zurich, Zurich, Switzerland. ✉e-mail: jbanfield@berkeley.edu; gavin.knott@monash.edu

sequences. Their linear genomes range up to 1.1 Mbp in length and are terminated by long inverted repeats. These features are shared by some plasmids[4], and it has been stated that Borgs are (obviously) plasmids. However, linear genomes are also typical of some viruses of archaea[5] and many giant eukaryotic viruses[6,7], previously termed nucleo-cytoplasmic large DNA viruses (NCLDV)[8]. Features of some *Nucleocytoviricota* that are reminiscent of Borgs include their very large genome sizes, substantial inventories of metabolic genes, and their propensity for gene acquisition[9]. Our recent study used in silico structure prediction to uncover the presence of some capsid-like proteins[3,10], raising the possibility that Borgs may be virus-like.

In this work, we prioritize multicopy protein families and apply in silico structure prediction to expand functional annotation across seven complete or near-complete Borg genomes and the full set of 17 Borgs, reasoning that the presence in multi-copy could indicate their particular importance. Prior works showed that Borg genomes encode a surprising inventory of metabolic and energy-relevant capacities, both as single genes and as multi-gene clusters[3]. Examples include four proteins of the methyl-coenzyme M reductase (MCR) complex central to methane metabolism, two proteins for biosynthesis of F430 (the cofactor for MCR), electron transfer via ubiquitous multiheme cytochromes, polyhydroxybutyrate production, and nitrogen fixation[3,10]. However, standard annotation methods predict functions for only ~20% of Borg proteins. Thus, we used in silico structure prediction to improve functional annotations of proteins from seven complete and essentially complete Borg genomes that we published recently[3,10]. We place the findings into context via an analysis of proteome content and organization of all 17 Borg genomes.

## Results

### Borg genomes encode multicopy proteins, many of which are related to deubiquitinases

Predicted structures of 8847 proteins were generated for the 9661 proteins of Orange, Black, Green, Amber, Amethyst, Cobalt and Ruby Borgs using AlphaFold2 in ColabFold[11–13]. These Borgs were selected from the previously reported 17 genomes that were used to define two major clades[10]. Structures were also generated for a subset of proteins from other Borgs (Supplementary Data 1A–C). Output models were assessed in terms of per amino acid prediction confidence (Supplementary Data 1D) and median pLDDT scores (Supplementary Data 2). In some cases, structures were also predicted using AlphaFold3[13]. Possible functions were suggested based on homology to existing experimental structures in the Protein Data Bank (PDB; Supplementary Data 3–8) using FoldSeek[14] or Dali[15]. Confident predictions (median pLDDT > 0.7) were generated for 69% of proteins, of which 36% had potentially informative hits in PDB (bitscores > 200; Supplementary Fig. 1). The median pLDDT score for these proteins was 93.92 (Supplementary Data 2). Notably, 48% of confidently predicted structures had bitscores of < 100 and 25% had bitscores of < 50, probably reflecting a lack of representation of many Borg proteins in the PDB (or substantial divergence). For this reason, even low bitscore matches were evaluated. Structural alignments were visualized in ChimeraX, and some extracted portions were realigned or reanalyzed to identify misoriented regions or extra domains.

We identified 28 Borg protein subfamilies (clusters of sequences with similarities high enough to suggest shared functions) that are particularly highly represented in the 17 genomes (Fig. 1A and Supplementary Data 9). The individual subfamilies were defined previously[10]. Most of the 1596 proteins in the 28 subfamilies had no, or

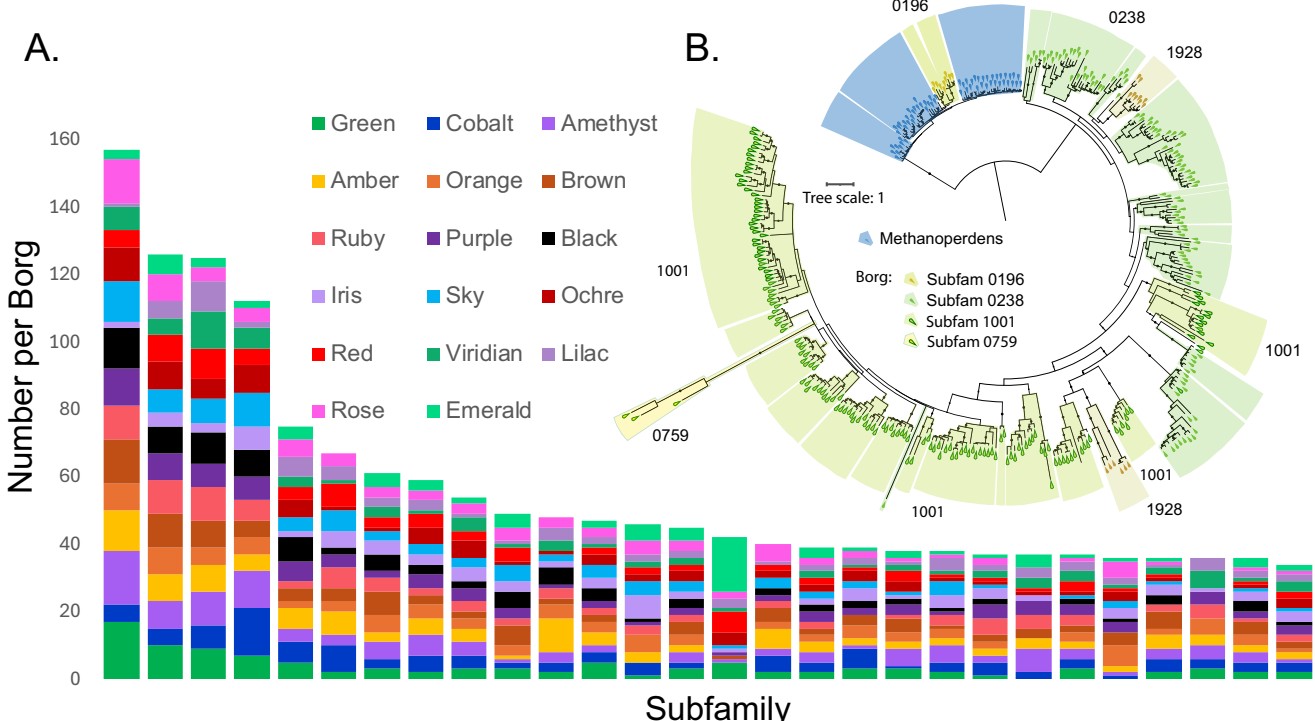

**Fig. 1 | Highly represented Borg protein subfamilies. A** Number of proteins assigned to each highly multicopy protein subfamily for each of the 17 Borg genomes based on clustering reported by Schoelmerich et al. 2024. Each distinct Borg has a color-based name. **B** Maximum-likelihood phylogenetic tree showing intermixing of Borg proteins from four of these subfamilies, supporting their treatment as a single group (Group 1). A few sequences without subfamily clusters were assigned to subfamilies based on phylogeny. Notably, subfamily 196 places within a large clade of *Methanoperedens* sequences, suggesting that Borgs acquired these 10 sequences by recent lateral transfer from *Methanoperedens*. Representatives of all subfamilies have the expected structures and active site residues (including 0759). A few proteins were too poorly folded to enable confident analysis, but in all cases, the protein sequences phylogenetically placed in clusters of proteins with the expected structures.

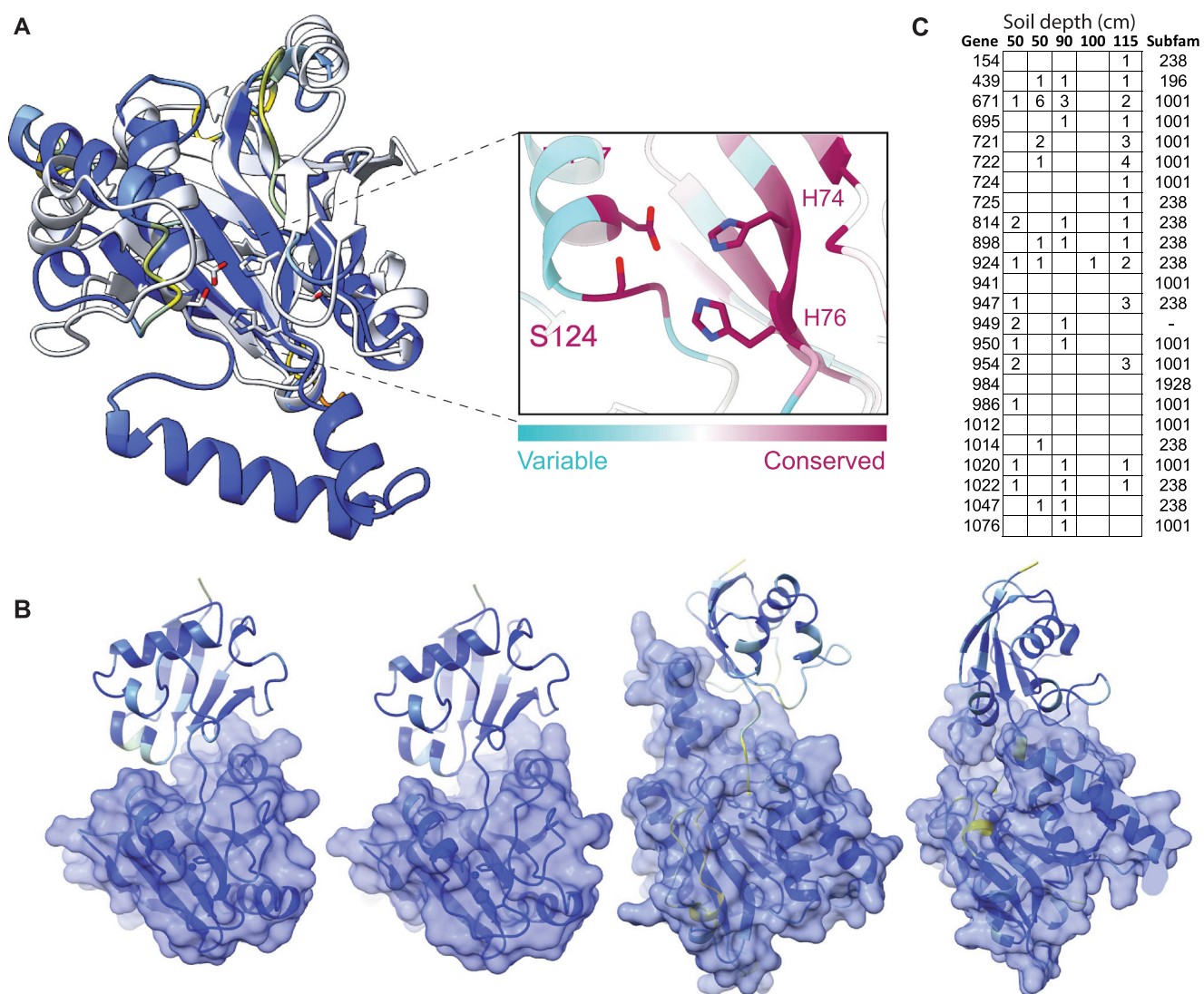

**Fig. 2 | Putative de-SAMPylation enzymes and gene expression data. A** Human AMSH deubiquitinase (PDB 2znr) and Borg deSAMPylase (Orange 866) colored by AlphaFold2 pLDDT confidence (blue is high confidence). The inset shows the conservation of the Borg protein active site based on a multiple sequence alignment of Borg proteins and the residues expected for Zn metalloproteases (Zn localization to the active site is well supported: ipTM: 0.97, PTM: 0.89). **B** AlphaFold3 predictions for cMp, and Orange Borg putative deSAMPylasess (surface representation from left to right: cMp 650, Orange 478, 878 and 883)

bound to *Methanoperedens* SAMP (cartoon) illustrating structural variability. Models are colored by sequence confidence (pLDDT), and the ipTM scores for the five models for each protein range from 0.81 ± 0.02 to 0.89 ± 0.01 (Supplementary Data 10). **C** Metatranscriptomic read counts (from a published nanopore dataset;[10]) for Black Borg de-SAMPylases in five soil samples and the subfamily affiliations of each protein. Expression was detected for up to 21 of the 24 proteins. The gene numbers reveal that these multicopy proteins are often encoded in close proximity.

only poorly defined, functions assigned using standard annotation methods. The most highly represented (subfam1001) had a few low-scoring structural matches to proteins in the PDB. The third most highly represented, subfam0238, had similar functional predictions, as did representatives of three much less commonly detected subfamilies (0196, 0759, 1928) and a few structurally similar proteins not assigned to any subfamily. The 315 proteins are phylogenetically intermixed (Fig. 1B and Supplementary Data 10), thus all were assigned to Group 1, with an average of 18.5 representatives per genome. The structure predicted using an alignment of the 315 protein sequences without using PDB templates was closely similar to that predicted using PDB templates and had similar fold confidence scores (Supplementary Fig. 2).

Group 1 proteins share a common fold with eukaryotic JAMM (AMSH) deubiquitinases (Fig. 2A). These Zn-metalloproteases liberate ubiquitin from target proteins using a highly conserved catalytic core comprising a nucleophilic Ser and metal ion-coordinating His/His/Asp or Glu residues[16] (Supplementary Information). Although deubiquitinases are known to remove ubiquitin from a variety of biomolecules, predicted cytoplasmic localization suggests that the targets are proteins. We mapped sequence conservation from the alignment of 315 Borg sequences (Supplementary Data 11A, B) onto the predicted structures and found that conservation localized to the expected active site, which likely (ipTM 0.97) binds Zn and is adjacent to the beta sheet face (Fig. 2A). Most of the 315 Borg proteins are likely metalloproteases with the expected active site residues (e.g., Supplementary Fig. 3A). The few with active site variants (e.g., Amber 1107 and Amber 1231, Supplementary Fig. 3B, C) are also confidently predicted to bind Zn but may have modified functionality.

Unlike Eukaryotes, Archaea generally do not use ubiquitin but instead post-translationally label their proteins using a Small Archaeal Modifier Protein (SAMP;[17]). Using the sequence of the characterized

*Haloferax* SAMP[18], we identified SAMP with the expected conserved β-grasp fold and di-glycine tail encoded in *Methanoperedens* genomes (e.g., cMp 2846) and in Lilac Borg (443). An E1 SAMP-adding enzyme is encoded in the cMp genome (3305) and co-folding confidently predicts it to bind the cMp SAMP within the active site (Supplementary Fig. 4A). *Methanoperedens* (Supplementary Fig. 4B) and Borg deSAMPylase-like proteins (Supplementary Fig. 4C) are predicted to form a similar interaction with cMp SAMP to position the di-glycine tail adjacent to the Zn bound active site. Multimer prediction datasets are provided in Supplementary Data 12. Across the diversity of putative Borg deSAMPylases, the multimer confidence values range up to ipTM = 0.9, pTM = 0.92, with strong model convergence (Supplementary Data 13), although many multimer predictions are not confident. The ipTM values are not predictable based on phylogenetic clade (Fig. 1B). Low SAMP interaction confidence may indicate divergent functions in some duplicated versions or an unconsidered factor, such as post-translational modification. A putative Lilac Borg deSAMPylase binds its own SAMP to the active site, but the score is below our confidence threshold (ipTM = 0.41, pTM = 0.72). Overall, the combination of JAMM protease-like fold, conserved active site residues, Zn coordination and a subset of multimers with confident SAMP binding indicates that Group 1 Borg proteins are Zn-metalloproteases, and many are likely deSAMPylases.

The complete ~ 4 Mbp *Methanoperedens* genome (cMP) has two putative deSAMPylases with the expected topology and conserved active site residues. cMP 1357 falls within a clade defined by sequences from *Methanoperedens* and other archaea, whereas cMP 650 falls within a clade of other *Methanoperedens* sequences that includes a subclade comprised of 10 Borg proteins (mostly subfam0196; Fig. 1A). The most likely interpretation is that 10 Borgs acquired these confidently predicted deSAMPylases (Supplementary Data 14) from *Methanoperedens* via (recent) lateral transfer. This reinforces the established pattern involving the sharing of proteins between Borgs and *Methanoperedens*[3], supporting their physical association.

An obvious question is why Borg genomes encode so many putative deSAMPylases. There is substantial predicted variation in the shape of the region that would bind the SAMPylated protein (or other molecule; Fig. 2B), so distinct variants may deSAMPylate different targets. Specificity may be enhanced by structural rigidity conferred by disulfide bonds distant from the active site that are predicted in almost all examples (often three or four cysteine pairs per protein). Supporting complementary functionality, available nanopore transcript data for Black Borg[10] show that multiple variants were expressed in situ (Fig. 2C). Recent results indicate that bacteria add ubiquitin to proteins to block phage packaging and interfere with reinfection[19], so deSAMPylation may be a response to an archaeal host SAMP-based defense mechanism.

Given the proliferation of putative Borg deSAMPylases and long branch lengths in the tree suggestive of rapid divergence, we suspected that these multicopy genes are under increased diversifying selection relative to those of *Methanoperedens*. The dN/dS ratio for the large Borg clade (Fig. 1B) is 0.235, compared to 0.195 for *Methanoperedens* and 0.172 for the Borg clade nested within the *Methanoperedens* group. Although none of these values is indicative of positive selection (dN/dS > 1), the use of dN/dS as a purely comparative metric suggests that selection is relaxed in genes of the large Borg clade. Using the RELAX function in Hyphy[20], we calculated a relaxation parameter, K, of 0.53, which was verified to fit the observed data with a p-value of 0.0000**. Therefore, we conclude that relaxation of negative selection is significant in the large Borg clade ($K < 1$, $p < 0.05$).

### Borg investments in glycoconjugates

Structural prediction revealed many multicopy genes with functions likely related to surface modification, particularly production of glycoconjugates. Roles in N-glycan biosynthesis are suggested for diverse

members of subfam0199 (Fig. 3A and Supplementary Data 14), whose predicted structures correspond well with those of dolichyl phosphate mannose synthase (DPMS; Fig. 3B). This protein catalyzes transfer of mannose from GDP-mannose to the dolichol carrier Dol-*P* (an isoprenoid) to yield dolichylphosphate mannose, likely for decoration of external proteins. The predicted surface representation of the Borg DPMS proteins reveals a volume that could accommodate the UDP-mannose donor substrate with the donor mannose group positioned proximal to the lipid entry channel (Supplementary Fig. 5A, B). Nanopore metatranscriptomic data[10] demonstrate expression of four of the five Black Borg DPMS proteins in wetland soil, consistent with complementary functionality.

The DxDxQ (particularly DxQ) residues of the metal binding motif required for DPMS functionality[21] occur in many, but not all, phylogenetically defined Borg clades, and in the DPMS from *Methanoperedens* and other archaea (Fig. 3A and Supplementary Information). There is variability in the number of alpha helices in the membrane anchor, and 13 of the 17 Borgs have at least one DPMS that completely lacks transmembrane alpha helices (Fig. 3C). Only members of one clade that lack the alpha helical anchor have the DxDxQ motif (Fig. 3A). Like all DPMS, anchorless variants have amphipathic helices with a hydrophobic face (Supplementary Fig. 5C) that likely enables localization at the cell membrane. Some anchorless variants are likely functional if they have the conserved DxDxQ motif, given the functionality of truncated DPMS from *Pyrococcus furiosus*[21]. However, the active site residues are different in some anchorless variants (Supplementary Fig. 5D).

Adding to DPMS diversity, five Borg and two archaeal proteins have an extra domain that aligns to the primary cytoplasmic domain of DPMS (Fig. 3D and Supplementary Fig. 6A, B). Alphafold sometimes predicted homo-tetrameric structures (Supplementary Fig. 6C, albeit without multimer confidence; Supplementary Data 12), recreating the known tetrameric eukaryotic homolog structure[22]. All seven proteins with an additional cytoplasmic subunit have metal binding sites in the main cytoplasmic subunit and two also feature the metal-binding DxDxQ motif in the extra domain (Supplementary Fig. 6D). One predicted protein with the DxDxQ motif is split (Brown 377,378), but it should be functional following a + 1 frameshift (Supplementary Fig. 7). Overall, the structural diversity hints at broad functionality that may be achieved via proteins with and without a membrane anchor, via a second cytoplasmic domain, and via multimerization as homo- or heteromers.

Consistent with the pattern for deSAMPylases, the Clade 7 Borg DPMS phylogeny (Fig. 3A) is suggestive of diversification following acquisition of Borg proteins via lateral gene transfer from *Methanoperedens* or related archaea. However, single *Methanoperedens* DPMS-like proteins within Clades 3 (6 Borg proteins) and 5 + 6 (14 Borg proteins) suggest that the reverse process, in which DPMS were acquired by *Methanoperedens* from Borgs, also likely occurs. The dN/dS values for Borg proteins = 0.121 compared to 0.098 for *Methanoperedens*, with a relaxation parameter, K, of 0.76, and $p = 0.0241$**, consistent with significant relaxation of negative selection in Borgs ($K < 1$, $p < 0.05$). Thus, it now appears that variants that undergo diversification in Borgs can make their way back into host genomes, contributing to host evolution.

Two DPMS genes occur in Cobalt Borg in a region that is enriched in genes seemingly involved in cell surface decoration (Fig. 4). Similar regions occur in all Borgs. The Cobalt Borg region encodes multiple UDP-sugar-epimerases, glycosyltransferase MshA, deacetylases, and genes that may be involved in the biosynthesis of mycothiol, a complex N-acetylglucosamine-based polymer. Also present is a gene cluster for which structural analysis supports a role in biosynthesis of the polysaccharide O-methyl phosphoramidate, which contains unusual C-P bonds, genes that may form phosphonates, and others with predicted roles in phospholipid and glycerophospholipid metabolism

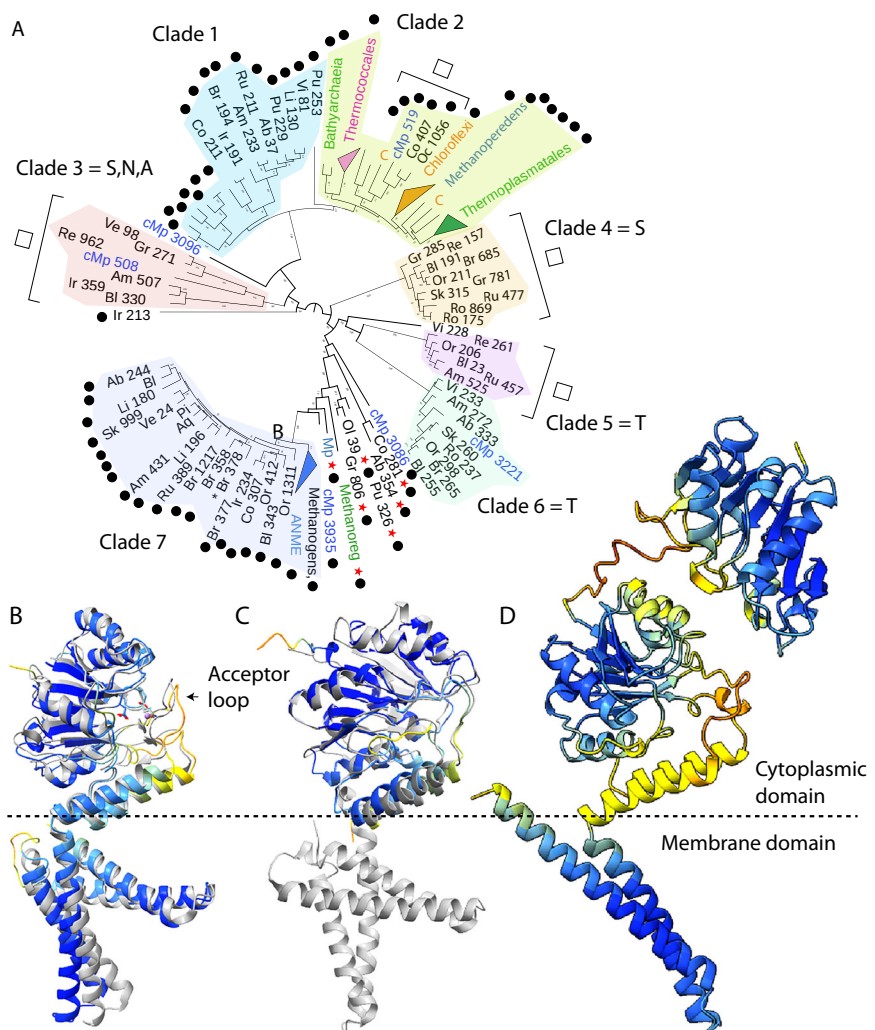

**Fig. 3 | Borgs encode multicopy proteins related to dolichyl phosphate mannose synthase, implicated in glycosylation. A** Phylogenetic tree of putative DPMS proteins (first two letters indicate the Borg name (except for Amber, Ab), followed by the gene number). Almost all unnamed sequences are from archaea. C = Chloroflexi, cMp = complete *Methanoperedens* genome. Black dots indicate DPMS with metal-binding residues. Residue(s) in place of glutamine are indicated in the clade name. Open boxes indicate groups that lack the membrane anchor, and red stars identify proteins with extra cytoplasmic domains. The asterisk indicates a split protein likely functional following +1 frameshift. Some proteins may have been transferred to Borgs from *Methanoperedens* (e.g., Clade 7) and others from Borgs back to the *Methanoperedens* (e.g., cMp in Clade 6). **B** Cobalt 211 aligns very well to PDB 5mm0, and has all 40 conserved residues identified of the reference structures. **C** Some putative DPMS, e.g., Orange 206, lack the membrane anchor, but align well in the cytoplasmic region. **D** Other putative DPMS, e.g., Green 806, have an extra cytoplasmic domain. In (**B**, **C**), PDB 5mm0 is gray, and in (**B**–**D**), the Borg proteins are colored by the AlphaFold plDDT scheme (darker blue indicates higher confidence).

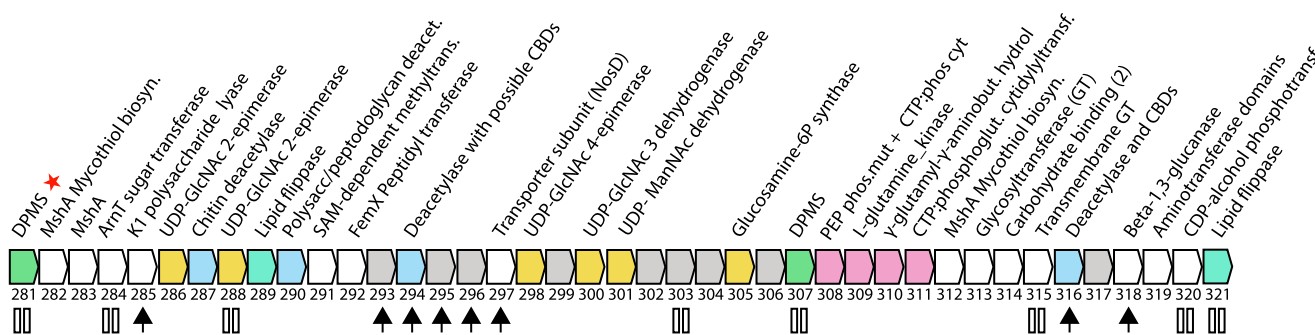

**Fig. 4 | Borg genomes encode regions enriched in proteins for production of glycoconjugates and other cell surface polymers.** A region of the Cobalt Borg genome (genes 281–321) encodes genes for N-glycan production (green, aqua; red star indicates that the DPMS has an extra cytoplasmic domain), with functions related to metabolism of UDP-N-acetylglucosamine (GlcNAc) and similar compounds (yellow), as well as deacetylases (blue) and proteins related to the production of O-methyl phosphoramidate (pink) (e.g., 311: CTP:phosphoglutamine cytidylyltransferase) and phosphonates (308 is likely a fusion of phosphoenolpyruvate mutase and CTP:phosphoglutamine cytidylyltransferase). CBD: carbohydrate binding domain. Paired vertical lines indicate transmembrane domains and arrows denote a SEC signal sequence.

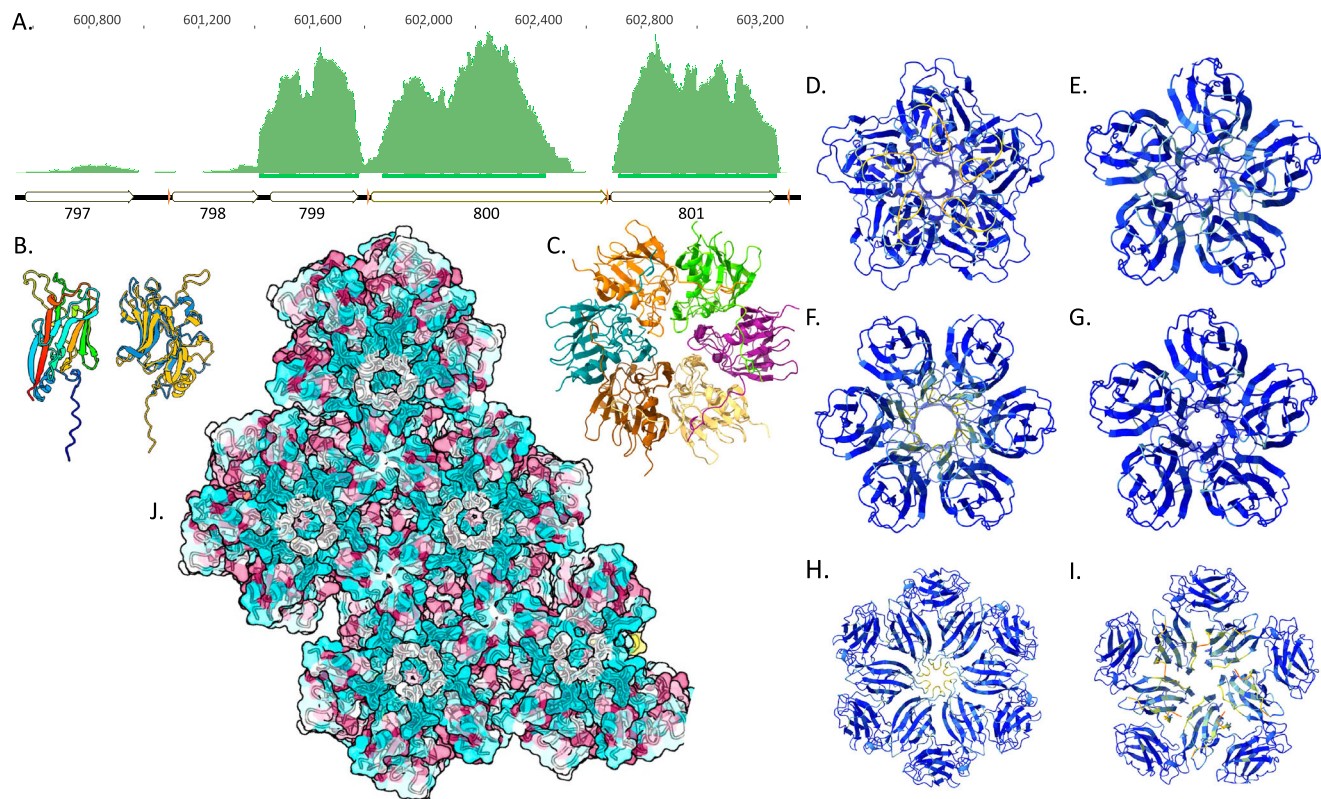

**Fig. 5 | Capsid-like proteins that multimerize into capsid-like arrays.**
**A** Metatranscriptomic reads mapped to the Black Borg genome primarily localize to three genes, the third of which is a putative capsid protein (801). Red tick marks are ribosome binding sites. **B** Black 801 monomer colored by position (start blue, end red) aligned to the reference 6h9c VP7 subunit M (gray). **C** Hexamer prediction for Black 801 showing the individual subunits in different colors (global ipTM = 0.8, range of all chain pair ipTM values is 0.66– 0.83, pTM = 0.82). Buried area between A and B subunits = 2564.9 Å². Model convergence RMSD for 5 models (pruned atom pairs): 0.284–0.361 Å. **D** Amethyst 1054 homopentamer, ipTM = 0.94– 0.95 for all five models, range of all chain pair ipTM values is 0.89– 0.92.; pTM = 0.94-0.95 for all five models. The homohexamer confidence values are similar: global ipTM = 0.91 for all models, the range of all chain pair ipTM values is 0.78–0.88. **E** Amethyst 1055 homopentamer, global ipTM = 0.91–0.93 for all five models; pTM = 0.92–0.93 for all five models. The homohexamer confidence values are similar: ipTM = 0.89–0.90. **F** Amethyst 1055 homohexamer, global ipTM = 0.89–0.9 for all

models, range of all chain pair ipTM values is 0.8 – 0.87; pTM = 0.9–0.91 for all five models. **G** Amethyst 1056 homopentamer, global ipTM = 0.95 - 0.96 for all five models, range of all chain pair ipTM values is 0.9 – 0.92; pTM = 0.95 - 0.96 for all five models. Only slightly lower values were achieved for homotetramers and homohexamers. **H** Amethyst 1057 homohexamer, global ipTM = 0.87 - 0.89 for all five models, range of all chain pair ipTM values is 0.73 - 0.9, pTM = 0.88 - 0.9 for all five models. Only slightly lower values were achieved for homopentamers. **I** Amethyst 1058 homopentamer, global ipTM = 0.86 - 0.87 for all five models, range of all chain pair ipTM values is 0.76 – 0.89; pTM = 0.88 - 0.89 for all five models. In (**E**–**J**) proteins are colored by AlphaFold2 pLDDT confidence scores. **J** One outcome for a prediction of the 24-mer of Black 801 showing assembly of hexamers into a capsid-like sheet, with sequence conservation (red) at the hexamer junctions. Note that the Predicted Aligned Error (PAE) data for multimers are reported in the .json files for each multimer.

(Supplementary Information). Eight of the forty Cobalt Borg proteins analyzed (Fig. 4) include SEC signal sequences (for localization and processing at the host membrane), and eight are predicted to be membrane-embedded. These findings suggest that Borgs invest in the production of diverse and unusual glycoconjugates, likely for surface decoration.

### Investment in encapsulation

Multicopy subfam2354 and subfam1011 (Fig. 1) are profiled as viral capsid proteins, paralleling a prior report of Borg proteins similar to those of an archaeal virus[10]. The capsid-like proteins are often most similar to those of *Haloarcula californiae* HCIV-1, an archaeal icosahedral virus with a linear genome, an internal membrane and a tailless icosahedral morphotype that includes hexamers of single jelly roll (SJR) major capsid proteins. However, as some SJR proteins have cellular functions, we compared the Borg SJR proteins to a selection of these, as well as to eukaryotic and archaea capsid proteins[23]. Only archaeal capsid proteins provide reasonable alignments (e.g., Black 801 to pDB 6h9c with a sequence alignment score of 71.7, RMSD over 66 pruned atom pairs of 1.109 Å; Supplementary Data 15).

In all 17 Borgs, a syntenic region encodes one capsid-like protein and either four or five hypothetical proteins. For Black Borg, the only Borg for which extensive transcript data were recovered from an Illumina dataset generated for an 80 cm depth soil sample, the SJR protein is 801. 801, 799 and 800 are exceedingly highly expressed (Fig. 5A); in fact, 76% of the Black Borg transcripts map to this locus (allowing 2% SNPs). The analogous Purple Borg genes are also relatively highly expressed. We now identify similar capsid-bearing gene clusters in some mini-Borgs (Borg-like ECEs with genomes ~10x smaller than Borgs;[24] (Supplementary Fig. 8), and some are expressed. Major capsid proteins are typically among the most highly expressed genes in viruses[25], supporting the inference that these proteins are involved in encapsulation.

We predicted a range of multimerization states for putative capsid-like proteins (Supplementary Data 12) and evaluated prediction confidence and model convergence (Supplementary Data 16). The putative capsid protein Black 801 (Fig. 5B) converges on predictions that assemble into homohexamers with high confidence (Fig. 5C). Scores for homopentamers are lower, and those for multimers with fewer or more subunits are much lower, indicating that hexameric and

possibly pentameric assembly is probable. We also analyzed a range of multimer compositions for Amethyst Borg capsid-like SJR proteins and found consistently very high ipTM values ($0.91 \pm 0.03$) and good model convergence for both homopentamers and homohexamers for four of five consecutive proteins (1054-1058; Fig. 5D–I). The strongest support is for homopentamer formation for 1056, with support for homotetramers and homohexamers as well. Only homopentamer formation is supported for 1058 (ipTM 0.96). Some of these proteins may be capable of forming both stable homopentamers and homohexamers, a feature of many icosahedral viruses[26]. Other icosahedral viruses (e.g., adenovirus) use two different proteins for the pentameric and hexameric capsomers, which is another possibility, given the multiplicity of capsid-like proteins[26].

In some simulations, rings composed of five or six monomers form extended sheet-like arrays, with conserved amino acids at interfaces between hexameric units, as expected for capsids (Fig. 5J and Supplementary Data 12). However, the ipTM values were low, and multimer conformations were inconsistent. We also demonstrated this for HCIV-1 major capsid proteins, aligning with the known limitations of AlphaFold for predicting higher order assemblies[27]. Overall, the extremely high level of expression and consistent homopentameric and/or homohexameric assembly (as known for HCIV-1) align with capsid-like behavior.

In addition to the region encoding 801, Black Borg also encodes five consecutive SJR-like proteins (867–871), and proteins with similar folds occur sequentially in 12 other Borg genomes (Supplementary Data 17). These findings indicate a surprising variety of potential capsid-like proteins in Borg genomes.

Some mini-Borgs also encode an additional SJR-like protein, and these define a large phylogenetic cluster that includes one Black Borg protein (867). Overall, 867 shares 69–80% amino acid identity with 20 different proteins from two different mini-Borg types, yet only 39% identity with the homolog from the most closely related Borg (Ocher). This provides evidence for the recent lateral transfer of capsid-like proteins between Borgs and mini-Borgs.

Eight Borgs have an additional gene cluster that typically encodes three proteins, two of which are nearly isostructural, and two genomes have duplicates of this region in close proximity (Supplementary Data 17). Structural features and distant homology identified by structure-based HMM model refinement (see "Methods") suggest tail- or capsid-like functions. This gene cluster and the two previously discussed gene clusters are typically encoded within an ~100 gene interval (Supplementary Data 17), suggesting consistent co-localization of functions likely related to encapsulation.

Other multicopy proteins encoded in all Borg genomes (subfam1773) resemble a bacteriophage HS1 tail needle knob protein (4k6b; Supplementary Fig. 9A). The Borg proteins have a SJR structure (Supplementary Fig. 9B) and display a hydrophobic alpha helical region that may insert into a membrane. Like 4k6b, they assemble into homo-trimers, but with moderate confidence (Supplementary Fig. 9C and Supplementary Data 17), and other multimers with much lower confidence, with conserved residues (Supplementary Data 18) at monomer junctions (Supplementary Fig. S9C). Alternatively, these may be capsid-related proteins.

## Borg characteristics resemble those of giant eukaryotic viruses
Given limited evidence for typical viral proteins other than those predicted to be capsid-like, we comprehensively compared Borg proteins with a large repository of viral sequences (BFVD,[28] but found no informative matches (no bitscores > 67). This simply may be a reflection of the novelty of Borg proteins. However, credible capsid-like proteins, the presence of viral-like (Herpesvirus) replication machinery[29], combined with linear genomes of large size, motivated a comparison of Borgs with a diverse set of *Nucleocytoviricota*.[30] Like Borg genomes, many linear *Nucleocytoviricota* genomes are terminated by long inverted repeats, yet their coding structure is different, as genes frequently alternate between strands. Similar to Borgs, many larger *Nucleocytoviricota* (e.g., Mimivirus and Megavirus) have lower %G + C content compared to their hosts[31]. *Nucleocytoviricota* exhibit high levels of horizontal gene transfer with other viruses[32] and organisms, a Borg feature that led to their naming. These acquired genes often come from the hosts, in the case of Borgs, from *Methanoperedens* archaea.

Arguably, the most striking feature of Borgs is their large inventory of genes that are normally only associated with organisms[3,10], (Fig. 6 and Supplementary Data 19), also true of *Nucleocytoviricota*[33]. As in *Nucleocytoviricota*[9], Borg inventories of sugar and lipid-related genes may be involved in decoration of capsid-like structures. In fact, given this, and their diverse inventory of putative capsid-like proteins, Borgs may construct capsids that are analogous to the complex structures reported recently for *Marseilleviridae*, which have eight protein components and an internal membrane, likely with a glycoprotein surface[33]. Like *Nucleocytoviricota*, Borgs encode genes for coenzymes, and nine Borgs have three sequential genes for the production of NAD. NAD production may counter host defense mechanisms that deplete cellular NAD levels[34]. Borgs and *Nucleocytoviricota* both encode genes for glycolysis / gluconeogenesis (in Borgs, up to seven genes, including for sequential steps encoded by sequential genes) and related genes of the pentose phosphate pathway. Other Borg proteins are involved in pyruvate, phosphoenolpyruvate, and acetyl-CoA metabolism, and the TCA cycle (e.g., sequential citrate synthase and aconitase genes, Supplementary Data 19, also see ref. 10). Notable in Borgs may be an extensive capacity to remove ubiquitin-like SAMP from proteins, paralleling the finding that some *Nucleocytoviricota* target host ubiquitination[35], in part to suppress host antiviral immune responses[36]. Thus, overall, investments in core carbon metabolic processes are features of both Borg and large *Nucleocytoviricota* genomes[37].

Unlike most *Nucleocytoviricota*, Borgs do not encode RNA polymerases, but they do encode many transcriptional factors to recruit host machinery to drive transcription. Both *Nucleocytoviricota*[38,39] and Borgs possess extensive translational machinery. For example, Borg genomes encode up to 23 tRNAs, 14 have one to three tRNA synthetases, they have one or two of four different ribosomal proteins and 15 encode a CCA-adding enzyme. Interestingly, similar genetic repertoires occur in some giant viruses of bacteria (megaphages;[40]. Like some *Nucleocytoviricota*, Borgs have genes for many other DNA and RNA functions, including for nucleotide biosynthesis. Similar to *Nucleocytoviricota*, they have genes for transport of K + [33] and other compounds[41]. Also paralleling *Nucleocytoviricota*[38], some Borgs have genes for radical defense (e.g., superoxide dismutase) and nucleic acid damage repair.

*Nucleocytoviricota* genomes typically encode DNA polymerase B[41]. All Borgs encode at least one DNA polymerase B (PolB). The Borgs PolBs group with processive rather than protein-primed PolBs. To our knowledge, processive PolBs are not known to be encoded in elements other than viruses. However, the Borg sequences do not place phylogenetically with those of Eukaryotic viruses, so a common ancestor of Borgs and *Nucleocytoviricota* is not phylogenetically supported (Supplementary Fig. S10 and Supplementary Data 20). Consistent with this distinction, Borg capsid-like proteins are most comparable with those of archaeal *Sphaerolipoviruses*, not the capsids of *Nucleocytoviricota*.

*Nucleocytoviricota* typically encode proteins to dynamically package DNA into chromatin[42,43]. Many Borgs have one or more histone proteins, histone remodeling helicases, histone protein methyl/ acetyl transferases, and histone protein demethyl/deacetyl transferases. Further underscoring an investment in high-order DNA compaction, Borgs all have a gene encoding 3'hExo ternary complex, an exonuclease that negatively regulates the abundance of histone mRNA

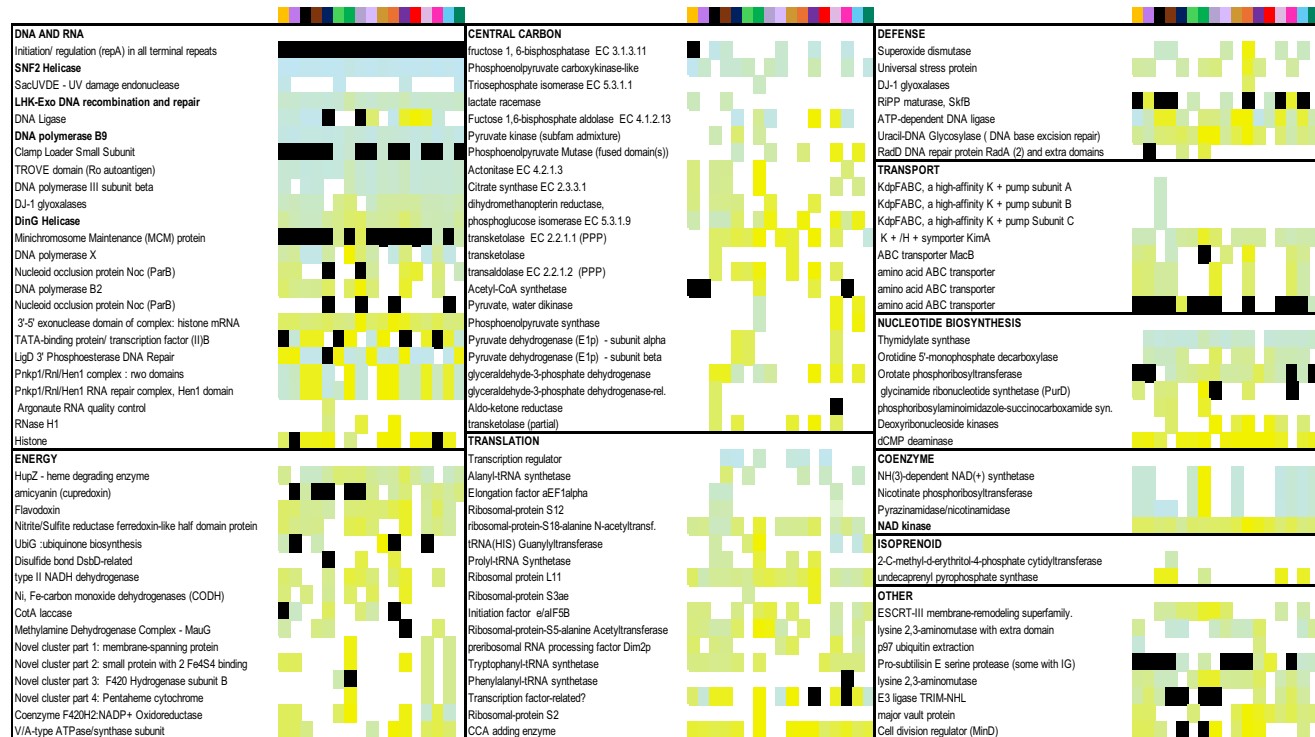

**Fig. 6 | Predicted protein functions across the 17 Borg genomes (columns ordered alphabetically), excluding the multicopy genes described elsewhere in the text and functions discussed in detail previously (e.g., multiheme cytochromes).** Rows are gene names, and genes are grouped by function type. Each column represents a Borg genome (colored boxes, top). Within each functional group, genes were sorted based on average position (approximately so as to not break up sequential genes). Genes in most of these categories occur in giant eukaryotic viruses. Genes that occur in multicopy in a genome are indicated in black (positions not indicated). Color indicates presence, and the shade of color indicates gene position in the genome, from early (blue) to late (gold); for details, see Supplementary Data 13. Although there is variation, genes for the same function tend to occur in similar genomic regions.

to appropriately titrate histone concentrations during DNA replication[44]. Interestingly, histone genes sometimes co-occur with a gene for a predicted vault protein (the subunits of which are predicted to assemble into vault-like arrangements). Less confidently functionally annotated are almost ubiquitous genes for a tubulin / FtsZ-like protein (Supplementary Information). Tubulin-like proteins occur in some *Nucleocytoviricota*[45] and in some bacteriophages, tubulin (PhuZ)-based spindles move the capsids to the surface of a phage nucleus for genome packaging[46]. Packaging ATPases similar to those used by *Nucleocytoviricota* could only be confidently identified in some mini-Borg genomes and one partial Apricot Borg genome (Supplementary Data 1C). There are multiple ATPase-like proteins in Borg genomes, some with structures resembling packaging ATPases, but their predicted functions vary. Given the high divergence of the capsid-like proteins to known capsids, it is perhaps not surprising that the packaging machinery would also bear only remote homology to similar enzymes. Thus, the presence of packaging machinery in Borg genomes remains unresolved.

Like *Nucleocytoviricota*[47], Borg genomes are peppered with genes for selfish genetic elements (e.g., transposons). In fact, genes for TnpB, IsrB and Cas12 are among the most prevalent multicopy proteins in Borg genomes (Supplementary Fig. 11). Similarly, many *Nucleocytoviricota*, especially those that integrate into the genomes of their hosts, also encode a variety of TnpB homologs called Fanzors[48].

Ubiquitous, tandemly repeated nucleotide sequences are striking features of Borg genomes (Fig. 7). These repeats are also prevalent in the wide diversity of *Nucleocytoviricota* (Table 1). Such repeats are very rare in the curated and complete *Methanoperedens* genomes, and occur ~ 3 to ~ 5 times less frequently (as microsatellites) in some bacterial genomes[49]. The tandem repeat patterns in *Nucleocytoviricota* and

Borgs are similar in terms of their distributions, the number of repeats per region, and average repeat length (Table 1 and Supplementary Data 19, 21–24). All but the shortest repeats are almost always novel to each region, and population-level variation in unit repeat number per locus in both is indicative of rapid evolution.

Of the tandem repeats, 44% and 39% occur within genes in Borgs and *Nucleocytoviricota*, respectively, and they almost always generate amino acid repeats (Table 1). In Borg genomes, amino acid repeats often introduce intrinsic disorder[50]. This is also suggested for some *Nucleocytoviricota* repeat proteins, but most are predicted to form ankyrin repeats or other alpha-solenoid structures[9]. Importantly, it is not uncommon for repetitive regions within proteins to be predicted as both intrinsically disordered and alpha-helical[51]. In ankyrin, the alpha solenoid structure is an emergent property of multiple intrinsically disordered repeats interacting with each other. In other cases, multimerization or post-translational modification causes low complexity regions to toggle between disordered and alpha-helical structures[52].

The ability to transition between disordered and ordered states is characteristic of proteins that undergo liquid-liquid phase separation to form separate cellular compartments that are not membrane-bound. Recent work has claimed that the viral factories[30] of the *Nucleocytoviricota* Mimivirus and Noumeavirus are phase separated organelles composed of, and scaffolded by, intrinsically disordered proteins[53]. Sequestration of viral replication also occurs through the production of a 'phage nucleus' in some jumbophage[54], possibly suggesting similar adaptations in large viruses. The high number of intrinsically disordered proteins in Borgs may hint at their ability to form phase-separated compartments similar to the *Nucleocytoviricota* viral factories.

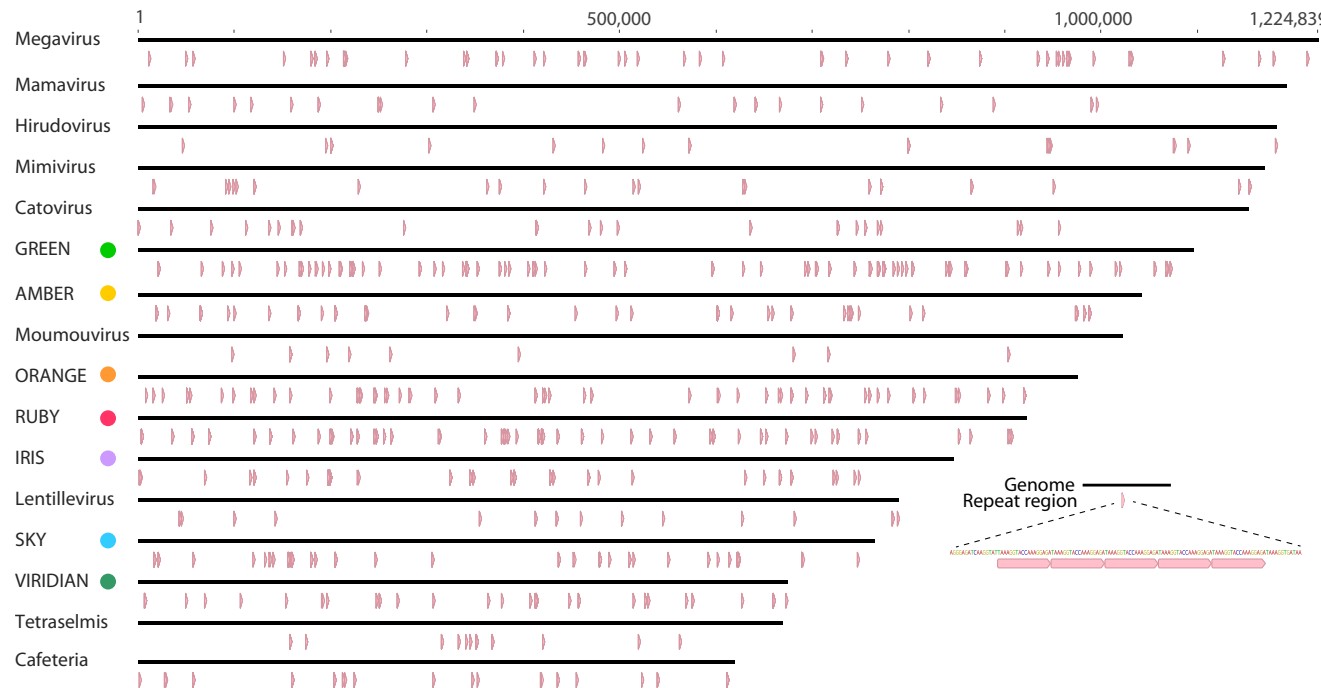

**Fig. 7 | Borgs and Nucleocytoviricota genomes encode tandemly repeated nucleotide sequences.** Borgs and *Nucleocytoviricota*, here using *Imitervirales* as examples, display generally similar abundances and distributions of perfect tandem repeats. Genomes for these and a selection of Borgs (colored dots), are listed in order of decreasing genome length. For more complete information for a wider diversity of viruses, see Table 1 and Supplementary Data 15–19.

**Table 1 | Overview of tandem repeats in Borgs, *Nucleocytoviricota* and the complete *Methanoperedens* (cMp) genome**

| | Genes with repeats | Intergenic with repeats | In genes, div by 3 | In genes, not div by 3 | Intergenic, div by 3 | Intergenic, not div by 3 |
|---|---|---|---|---|---|---|
| Borgs | 44 % (332) | 56 % (425) | 99 % (330) | 1% (2) | 24 % (103) | 76% (322) |
| Nucleocytoviricota | 39 % (906) | 61% (1415) | 97% (880) | 3% (26) | 53% (756) | 47% (659) |
| *Methanoperedens* | 50% (1) | 50 % (1) | 100% (1) | 0% (0) | 0% (0) | 100% (1) |
| | Sum genome length bp | Regions / 100,000 bp | Repeats / region | Av. repeat length bp | – | – |
| Borg (17 genomes) | 14525135 | 5.1 ± 1.8 | 5.4 ± 3.7 | 24.8 ± 16.3 | – | – |
| Nucleocytoviricota (65 > 500 kbp) | 77234387 | 2.9 ± 2.2 | 4.5 ± 3.3 | 25.3 ± 20.9 | – | – |
| Nucleocytoviricota (10 > 500 kbp) | 10923403 | 4.6 ± 4.0 | – | – | – | – |
| Nucleocytoviricota (1464 > 10 kbp) | 91899808 | 7.3 ± 5.7 | – | – | – | – |
| *Methanoperedens* (1) | 4003972 | 0.05 | 3 ± 0 | 47.5 ± 3/5 | – | – |

Tandem repeat regions were inventoried if they were at least 50 bp long with at least 3 repeat units and a minimum unit repeat length of 9 bp. Statistics are provided for a select group of Nucleocytoviricota with genome sizes of comparable size to Borgs (65, mostly Imitervirales), a more representative set of Nucleocytoviricota with large genomes, and a large set of Nucleocytoviricota with genomes >10 kbp in length.

## Clues to borg evolution

Borgs genomes feature many multicopy genes, and these are generally distributed throughout the genomes. The highly prevalent putative deSAMPylases are often encoded in genomic proximity, suggesting origination by gene duplication (Supplementary Data 19 and Supplementary Data 25). Other multicopy functions relate to protein decoration, nucleic acid manipulation, encapsulation and redox activities (Supplementary Fig. S11), and some of these occur as sequential copies in at least four of the seven Borg genomes (Supplementary Data 20). Multicopy genes also make up a large proportion of *Nucleocytoviricota* gene content[9]. In fact, protein duplication is a suggested driver of viral gigantism[38,55]. In *Nucleocytoviricota*, the "genomic accordion" process is suggested to maintain or expand host range[39,56,57]. Should the

evolutionary model of giant size through gene accretion extend to Borgs, mini-Borgs (with their ~ 10X smaller genomes[24] may resemble an ancestral state.

We noted consistent genomic locations for many genes that occur in two or more Borg genomes and are single copy per Borg genome (1931 examples). For example, DNA polymerase B9 family genes are all encoded early ($0.08 ± 0.2$ along the genome), and the widespread CCA-adding enzymes are consistently encoded ~ 75% of the way along the genome ($0.73 ± 0.11$). In fact, 64% of the 1931 genes occur at the same average position with a standard deviation of ≤ 0.1 (**Supplementary Fig. 12**), and 251 occur in such close proximity in 7 or more Borgs (Fig. 6 and Supplementary Data 19). This greatly extends the previously noted pattern of 40 universally present single-copy Borgs

genes in generally similar genomic positions[10]. Often, but certainly not always, the phylogeny of genes in similar genomic locations matches the phylogeny based on the 40 universal single-copy proteins. The findings indicate extensive inheritance of a genetic backbone, shaped by gene loss.

The presence of many of the currently existing Borg genes in a common Borg ancestor may align with the speculation that the functional gene-rich Borg backbone was acquired from an archaeon that shared a common ancestor with *Methanoperedens*[10]. It has been similarly suggested that giant eukaryotic *Nucleocytoviricota* arose via gene loss from an organismal source[38], but most phylogenetic evidence tends to support the evolution of viral gigantism from smaller viruses[8,58,59]. Alternatively, it is possible that the current pattern of gene position conservation and sequence reflects inheritance from a gene-rich ancestral ECE[8,60], possibly one that evolved at the time of origination of the *Methanoperedens* genus, with subsequent lineage-specific gene gain and loss. Regardless of the details, it appears that the last common Borg ancestor had a giant genome.

So, are Borgs giant archaeal viruses? They are undoubtedly archaeal, which makes the question: are Borgs viruses? Periodic viral-like bursts are suggested based on very high, and highly variable, Borg to host chromosomal DNA copy numbers in some samples. These phenomena may also reflect the selective preservation of encapsulated Borg DNA[10]. The small number of genes encoded within the known archaeal viruses precludes a detailed comparison with Borg gene content, and the lack of a ubiquitous packaging ATPase and for evidence of full capsid assembly limits the argument for classification of Borgs as viruses. However, Borgs encode functions that are similar to those of viruses in general, such as an SNF family helicase (possibly involved in chromatin remodeling), proliferating cell nuclear antigen (sliding clamp DNA polymerase), and DNA polymerase B, and the genomic placements are generally analogous. The DNA polymerase B genes are encoded very early in the large replichore, whereas genes involved in translation and ~100 kbp regions that encode many putative structural genes occur mid-genome. Histones that are probably required for organization and packaging of DNA (Supplementary Fig. 13, predicted heterodimer of adjacent encoded Amethyst Borg histone-like proteins have ipTM values of ~0.86 for all models) are encoded towards the end of the large replichore or within the small replichore (Supplementary Data 19). ESCRT-III-like proteins are present in some Borg (Supplementary Fig. 14 and Supplementary Data 27). These are involved in membrane remodeling, and in *Nucleocytoviricota* generate the lipid component of capsids.

Although there is likely a continuum of gene content between plasmids and viruses, Borgs apparently lack conjugation machinery, ParAB segregation systems and replication modules typical of plasmids[3]. No new plasmid marker genes came to light in the current study. Recently, we reported plasmids of *Methanoperedens* that have < 200 kbp circular genomes that mostly encode proteins involved in nucleic acid manipulation and a few translation genes that are not found in the host. Unlike Borgs, the plasmids occur in a similar copy number to the host and share little gene content with Borgs[61]. However, Borgs have numerous features suggestive of lifestyles somewhat analogous to those of giant eukaryotic viruses, and we conclude that the weight of evidence now supports a more viral-like than plasmid-like lifestyle. Regardless of the classification of Borgs, the parallels with *Nucleocytoviricota* are striking. Given no phylogenetic evidence for common ancestry, many *Nucleocytoviricota* and Borgs similarities are likely due to convergent evolution. Gigantism in eukaryotic viruses evolved independently in separate lineages[58,62], likely occurred repeatedly in bacteriophages to generate the many distinct clades[40], and may have occurred again in virus-like Borgs of archaea.

Koonin et al. write "In a sense, all evolution of life is a history of coevolution between MGEs and their cellular hosts"[63]. Several studies have suggested that ancient eukaryote-virus gene exchange shaped the early evolution of complex cells, possibly even leading to traits such as linear chromosomes or the nucleus[64,65]. Direct evidence for early virus-to-eukaryote gene exchange has remained elusive, and an archaeal virus that could serve as a clear progenitor in these evolutionary scenarios has not been found. It is intriguing to consider that archaeal evolution at around the time of eukaryogenesis may have been shaped by large and complex viruses. Given that Borgs lack clear phylogenetic affinity for eukaryotic viruses and eukaryotic replisome components, it is unlikely that they represent the missing link in early eukaryotic evolution. However, Saturn mini-borgs encode a delta-like family B DNA polymerase that has phylogenetic placement proximal to eukaryotic polymerases alpha, zeta, and delta (Supplementary Fig. 11 and Supplementary Data 20). This raises the possibility that these elements may have played a role in the early emergence of the eukaryotic replisome. Regardless, the existence of Borgs and mini-borgs opens the door to future work on other lineages of mobile elements and complex archaeal viruses that may have played a role in the emergence of cellular complexity.

## Methods

### Identification and analysis of multicopy subfamily proteins

Multicopy subfamily analysis was conducted by analysis of the subfamilies reported for the 17 complete and highly curated, near-complete Borg genomes that we reported, see (Schoelmerich et al. 2024. Subfamilies (Supplementary Data 26) for which functional and structural predictions were similar or the same (see below) were evaluated using phylogenetic analysis, and categories grouped as appropriate. Analyses did not rely upon strict subfamily assignments because structural alignments demonstrated similar folds for sequences from different subfamilies (and vice versa).

### Protein structure prediction and functional analysis

The genomes of Orange, Black, Green, Amber, Amethyst, Cobalt and Ruby Borgs were selected as representatives of Borg diversity based on their relatedness, as inferred from the phylogeny based on 40 single-copy universal genes that we reported previously[10]. For each protein in these seven Borg genomes, ColabFold (--amber --templates --num-recycle 3)[12] run locally was used to predict AlphaFold2 structures that were evaluated initially for similarity to proteins in the Protein Database (PDB, https://www.rcsb.org/). In total, we folded 8847 Borg proteins as monomers. The mean and median pLDDT values for all reported structures and other statistics are provided in Supplementary Data 1 and the median pLDDT is reported protein by protein for each of the seven Borg genomes. Information is also found in each.pdb file and data folder for each protein (Supplementary Information). We prioritized analysis of structures for proteins from subfamilies that occurred multiple times in the Borg genomes (multicopy subfamilies) and selected the most prevalent multicopy subfamilies for detailed analysis. Phylogenetic analyses (see below) were performed for the largest subfamilies and related subfamilies to distinguish clades and group subfamilies with essentially the same structures and inferred functions.

To obtain potential functional insights for proteins from multicopy subfamilies, we identified the structure model with the highest score (model wide pLDDT) to search against the Protein Data Bank (PDB) for structural similarity using Foldseek (https://search.foldseek.com/search). Most models selected for further consideration had a median model wide pLDDT of ~70, a value that meets the accepted definition of a high quality structure[11]. However, we did not limit our analyses to these models as median scores can be strongly influenced by local disordered regions (e.g., intrinsically disordered regions that are prevalent in Borg proteins[50]), yet other parts of the structures can provide functional clues. We chose to use the median rather than the mean value of pLDDT to reduce this effect. We favored a more inclusive approach, given the likely novelty and wide divergence of Borg compared to experimentally studied proteins.

To robustly assess proteins with potentially credible hits in the PDB, we visualized and manually investigated structural predictions for proteins representative of each protein subfamily discussed in this manuscript using UCSF ChimeraX (version 1.8, https://www.rbvi.ucsf.edu/chimerax). Although we prioritized structural predictions with bitscore values to a PDB structure of >200, subfamilies with no relatively high-scoring matches to a PDB entry were also evaluated, so long as the bitscores were >~50. These (and some other) models were searched against the AlphaFold database that includes predicted as well as experimentally validated structures using Foldseek. Analyses included information such as the local per-residue confidence (pLDDT) and agreement between local and global structural features of the PDB hit(s), as well as the distribution of charged and hydrophobic residues and amino acids of interest (e.g., associated with active sites). As we focused on multi-copy proteins, visualization also included residue by residue sequence conservation that was rendered from protein sequence alignments. In cases where portions of the protein were apparently rotated relative to the PDB reference, that portion was extracted and realigned. Unexpected regions (e.g., extra domains) were extracted, and the structures compared to existing domains (in cases where domains were duplicated) and to public data using Foldseek.

For monomers, we investigated the convergence of the five models (rank 0 - rank 4) by comparing the pLDDT values for the models generated using Colabfold. We concluded that AlphaFold reached convergence if all five models converge on the same structural solution. To evaluate this, we used TM-align[66] for all pairwise comparisons (335,911 comparisons). When requiring 100% alignment over the entire length, 91,677 were essentially identical (TM score 0.5 - 1 is considered essentially the same score). 183,377 proteins had 80% alignment and 80% identity.

For multimer predictions, we used AlphaFold3[13] with default settings and assessed the pLDDT, PTM (overall model reliability, derived from the predicted alignment error, PAE) and iPTM (interface pTM, the accuracy of the predicted relative positions of the subunits forming the protein-protein complex). For analysis of capsid-like proteins, multimers of 2 - 9 subunits were evaluated based on iPTM scores and the highest-scoring multimer was chosen as the most likely state. The iPTM scores >0.8 were considered confident, following AlphaFold guidelines. We analyzed the highest confidence prediction for each model using ChimeraX to visualize the local per-residue confidence (pLDDT), sequence conservation based on sequence alignments, and agreement between local and global structural features of the PDB hit. We assessed convergence of the five models, requiring similar high (>~80) iPTM scores for all five of the models generated after three recycles. Buried areas between multimer domains were estimated as a measure of interaction confidence using Chimera X. For putative deSAMPylases, complexes with SAMP were predicted from representatives of the major clades defined by phylogenetic analysis of protein sequences.

To test whether multimer prediction confidence could be improved for a putative deSAMPYlase complexed to SAMP, we augmented the ColabFold AlphaFold3 prediction using an MSA that included all Borg sequences and sequences from *Methanoperedens* and other archaea and an MSA for credible SAMP proteins from similar sources (Supplementary Data 11B). Calculations were performed with up to 100 recycles. As the difference between 20 and 100 recycles was nominal and no improvement was achieved, we did not perform an analogous analyses for all of the SAMP-deSAMPylase multimers.

Protein functional predictions also made use of domain identification tools (e.g., HMMer, https://www.ebi.ac.uk/Tools/hmmer/search/hmmscan), HMM-based analyses and Pfam profiles,[67] In a few cases where there was a sufficient number of reasonably similar Borg proteins available (ideal for multicopy proteins) and structure

predictions had low confidence and/or there was little or not detectably structural similarity, protein structure predictions were generated with AlphaFold2 using an input protein multi-sequence alignment without the use of PDB templates. Multisequence alignments were generated using MAFFT[68]. Some proteins were also analyzed using ESMfold (https://esmatlas.com/resources?action=fold) and Dali (http://ekhidna2.biocenter.helsinki.fi/dali/). Structural details, including active site residues, were analyzed by reference to the published literature. Transmembrane regions of proteins were predicted using DeepTMHMM[69] and signal peptides were predicted using SignalP6.0[70].

## Phylogenetic analysis of individual protein sequences
Dolichyl phosphate mannose synthase tree
DPMS sequences (subfam0199) from Borgs were aligned using MAFFT[68]. and manually refined. The final alignment contained 717 positions. Phylogenetic analysis was performed using IQ-TREE[71] with automatic model selection and 1000 bootstrap replicates. Trees were visualized using Geneious Prime 2024.0.4 (https://www.geneious.com) and annotated in Illustrator.

B-family DNA polymerase tree
We manually curated B-family DNA polymerases from Borgs, mini-Borgs, bacteria, archaea, and eukaryotic viruses, combining them with reference sequences from ref. 72. Sequences were aligned using MAFFT v7.490[68] with the L-INS-i algorithm. The alignment was trimmed using trimAL v1.4.rev15[73] with the 'gappyout' option. The final alignment of 646 positions was subjected to maximum-likelihood analysis using IQ-TREE v2.1.3[71]. The LG + C60 + R + F model was selected based on the Bayesian Information Criterion (BIC). Branch support was assessed using 1000 ultrafast bootstrap replicates.

deSAMPylse-like metalloprotease tree
The collection of putative deSAMPylse-like metalloproteases was established by combining the sequences from Borgs subfamilies assigned to Group 1 with sequences from the complete cMp *Methanoperedens* genome and related sequence recovered via BLASTP from NCBI. Sequences were aligned using MAFFT-LINSi and trimmed using trimAL with the 'gappyout' option, resulting in an alignment length of 171 positions. Phylogenetic analysis was conducted using IQ-TREE with the LG + F + G4 model and 1000 bootstrap replicates.

## Repeat analysis
A custom script (github.com/rohansachdeva/assembly_repeats) for repeat finding and visualization that was reported previously[50] was modified to report the coding and non-coding localization of repeats. Repeats were considered coding if they are fully contained within a coding region.

## Structure-based HMM model refinement
To attribute functions to previously unannotated proteins, we employed a structure-based method for the refinement of profile HMMs. We first used BLASTp to retrieve all protein homologs from Borg genomes. We then aligned these sequences using MAFFT v7.490, and manually curated the alignment by removing unaligned residues to restrict the alignment to the core protein domain. We then used HMMer to create a starting profile HMM, which we then used to search the UniprotKB database using HMMsearch[74]. Hits from this search were then added to the alignment. We then used AlphaFold 2 to predict the structure of the lowest scoring hit above the inclusion threshold[75], and then searched the structure against AlphafoldDB and the PDB using FoldSeek[14]. We added hits from this search to the alignment, which we then used to create a new profile HMM. This process was repeated until no new sequences could be added to the alignment. We then analyzed the functional annotations of sequences in the alignment and assigned putative functions to the original proteins.

## Transcriptomic analysis

Total RNA was extracted from ~ 5 g wetland soil samples using the Qiagen RNeasy PowerSoil Total RNA Kit. Ribosomal RNA (rRNA) was depleted using the NEBNext rRNA Depletion Kit, followed by a library construction using the NEBNext Ultra II Directional RNA Library Prep Kit. The RNA library was sequenced on the Illumina NovaSeq6000 PE150 platform in Maryland Genomics to generate metatranscriptomic data.

Raw sequencing data was processed using BBDuk (https://jgi.doe.gov/data-and-tools/software-tools/bbtools/) to remove low-quality reads. Putative rRNA reads were filtered by SortMeRNA (v4.3.6)[76]. mRNA reads were mapped to genomes using bbmap (https://jgi.doe.gov/data-and-tools/software-tools/bbtools/) with a minimum ident of 97% to calculate the transcriptional activities of genes.

## Reporting summary

Further information on research design is available in the Nature Portfolio Reporting Summary linked to this article.

## Data availability

As noted in Schoelmerich et al. 2024, the 17 *Borg* and *Methanoperedens* genomes referenced in this study are available via ggKbase at https://ggkbase.berkeley.edu/borgs_mp_nanopore/organisms, and have been deposited in the NCBI database under BioProject accession PRJNA1119519. Source data underlying all main and extended data figures are provided within the Supplementary Data files and/or at Zenodo. In total, 27 Supplementary Data items accompany this paper. Supplementary Data 1, 10, 11, 12 and 14 are available at Zenodo (https://zenodo.org/records/15795806); the remaining items (Supplementary Data 2–9, 13, 15–28) are supplied with the article (Excel tables and associated data files).

## Code availability

The code for the tandem repeat finder is available on GitHub: github.com/rohansachdeva/assembly_repeats

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

## Acknowledgements

Innumerable people provided input, but specific thanks to Drs. Ben Adler, Brady Cress, Rotem Sorek and David Komander. Dr. Mart Krupovic and Owen Tuck are thanked for thoughtful comments on the manuscript. We thank Dr. Tom Delmont and three anonymous reviewers for their inputs. L.V.A. support was provided by a UC Berkeley dissertation year fellowship. Funding to FOA and ZKB was provided by a National Institutes of Health grant (no. 1R35GM147290-01), to FC by the Australian Research Council (DP21010338, DP230101879), and to G.J.K. by the Snow Medical Research Foundation (SMRF2021-276). Snow Medical were not involved in the design of the study, data collection, analysis, interpretation of the data, the writing, or the decision to submit the article for publication. Funding was also provided by the Bill and Melinda Gates 901 Foundation (grant number INV-037174 to J.F.B.). The findings and conclusions are those of the authors and do not necessarily reflect

positions or policies of the Bill and Melinda Gates Foundation. Funding was also provided by the Chan Zuckerberg Initiative (to J.F.B.).

## Author contributions

J.F.B. designed the study. Protein subfamily and detailed protein structure analyses were performed by J.F.B., with substantial input from G.J.K. and specific input from F.C. (capsid-like proteins) and R.B. (glycoconjugates). J.F.B. generated the sequence collections with assistance from S.L. and R.S. and input from F.A., Z.B. and L.A. G.J.K. performed most analyses of histone-associated proteins, IsrB, TnpB, and Cas12f. R.S. performed most model convergence and the BFVD analysis. LEVA made the final versions of the phylogenetic trees. C.R. performed the dN/dS analysis and the structure-informed HMM search to uncover possible phage structural proteins. L-DS generated and analyzed new transcriptomic data, identified putative packaging genes, and refined the synteny analysis for one region by adding Mini-Borg data. M.S. provided input into analyses. Repeat analyses were performed by R.S., who wrote the script, and J.F.B., who analyzed the results. J.F.B. prepared the figures and wrote the manuscript, with substantial input from G.J.K. and specific input related to *Nucleocytoviricota* by F.A., Z.B., and F.C. The manuscript was revised based on comments from all co-authors.

## Competing interests

The authors declare no competing interests.
