## [Transparent Peer Review file · Nature Communications]

Convergent evolution of viral-like Borg archaeal extrachromosomal elements and giant eukaryotic viruses

Corresponding Author: Professor Jillian Banfield

Version 0:

Reviewer comments:

Reviewer #1

(Remarks to the Author)

This manuscript analyzes the proteomes of giant extrachromosomal elements—Borgs—using large-scale in silico protein structure prediction (AlphaFold2/3). By clustering thousands of Borg proteins into subfamilies, the authors identify repeated functional themes, including metabolic enzymes, deubiquitinase-like (deSAMPylase) metalloproteases, and capsid-like proteins. The study builds a case that Borgs may share features with large dsDNA viruses (Nucleocytoviricota-like), prompting discussion of whether Borgs should be classified as giant archaeal viruses or remain viewed as plasmid-like elements.

Overall, the manuscript provides a valuable resource by applying large-scale AlphaFold predictions to an extraordinary set of archaeal extrachromosomal elements.

Our central criticisms revolve around clarifying how confident the authors can be about predictions by establishing thresholds for the used scores, specific multimeric states (especially putative capsid proteins), and deeper mechanistic/structural details for ambiguous cases (e.g., metalloproteases). Addressing these points should significantly strengthen the manuscript.

Major Points

Thresholds for prediction scores

Monomers:

The manuscript cites pLDDT >70 as “confident” but does not clearly indicate whether this refers to a global (average) value or per-residue cutoffs. Likewise, pTM and ipTM thresholds are presented as indicative of confidence, but specific threshold validation is missing.

The authors should clarify how per-residue pLDDT scores were aggregated to define a protein as “confident,” whether local regions must meet the same threshold, and how this was weighted against the pTM score. A clear thresholding pipeline needs to be defined here, and statistics of how many proteins passed thresholds are needed per software used (see below).

Multimers:

The authors need to verify their ipTM thresholds, especially on archaeal protein datasets, and distinguish between AlphaFold 2 and 3 as their score thresholds will likely differ.

Model convergence

The authors specify that a limited number of three recycles are run in ColabFold. While sufficient for easy-to-predict folds, especially complex or multi-domain proteins, 3 recycles may or may not be sufficient for models to converge. This can be typically seen by not converging pLDDT plots for the five models. The authors need to define thresholds for model convergence and what they did with predictions that were not sufficient (e.g. running more recycles).

Additionally, browsing through the pLDDT plots of the predictions, we have a hard time following the argument that about 60% of predictions were high quality by evaluating model convergence and overall pLDDT distribution.

Use of AF vs. Colabfold

The authors used a mixture of prediction software. Scores between models cannot directly be compared, and they perform differently, especially in hard-to-predict cases where MSAs are thin. The authors need to state a clear strategy for which protein and which software was used. E.g., using vanilla AlphaFold 2 when Colabfold gave low-confidence predictions due to sparser MSA.

Moreover, large parts of the Methods section on structure prediction are ambiguous. The authors again need to develop a clear strategy with thresholds and a data management solution to share the data with all required metadata.

Structural similarity search

It is unclear why only searches against the PDB were performed for most predictions. This should be expanded to the full AFDB as well as against the BFVD from the Steinegger lab (<https://bfvd.foldseek.com/>) to illuminate viral similarity.

Accessibility of predictions

All prediction data needs to be available. Currently, only PDBs and pLDDT files are shared but PAE plots and MSAs are missing, as well as all JSON output files. This is absolutely crucial to evaluate prediction quality and to figure out what settings were used. We suggest organizing them in folders per protein and not folders for each data type as done now.

Bitscore Cutoffs (>200) and E-Value Thresholds in FoldSeek/Dali

The authors occasionally accept matches in the 50–200 bitscore range when no better hits were found. While that can be reasonable for a domain of unknown function, it is inherently more error-prone. For these borderline scores, it would be prudent to confirm via domain-by-domain alignment or partial alignments (e.g., discarding disordered N/C-terminal segments) to ensure the match is *truly* to the same fold. A more explicit mention of “secondary checks” for borderline structural similarities would help readers judge the reliability of each functional claim.

Confidence in Hexameric Capsid Proteins (Figures 5C, S9)

The manuscript relies on a single global ipTM (or pTM) to claim that its predicted capsid proteins form hexameric assemblies. However, simply reporting one global metric does not confirm a stable interface for every subunit pair, nor does it clarify whether other lower-order oligomers (e.g., dimer, trimer, pentamer) might also form. In many icosahedral viruses, pentameric as well as hexameric capsomers are required, and the lack of high-confidence pentamer predictions may cast doubt on the capsid hypothesis.

We strongly suggest providing pairwise ipTM Data and full PAE plots for all assemblies and illustrating local interface predictions to confirm a single stable complex rather than partial interactions for all shown multimer predictions (not just potential capsid proteins) in the manuscript. This will allow readers to evaluate whether the predicted subunit–subunit contacts are uniformly well-resolved rather than relying only on a single ipTM for the entire complex. We also strongly suggest attempting predictions for dimer, trimer, pentamer, and hexamer assemblies. If smaller (or pentameric) stoichiometries show very low confidence, the classification of these proteins as bona fide icosahedral capsid proteins remains uncertain. In other words, the lack of stable pentameric assemblies in particular may suggest an alternative biological function or a less conventional capsid organization.

Potential metalloproteases (Figure S3)

The study references Zn-metalloprotease functionality but does not explicitly show predictions with Zn. The authors should explicitly run AlphaFold3 (or equivalent pipeline) with Zn to test whether the predicted active site geometry is consistent with Zn binding. Also the catalytic triad or metal-binding site must have high local pLDDT consistently, or else structural claims about catalytic residues (also in the putative deSAMPylases) may be on shaky ground.

Standard Annotation Methods

The manuscript’s phrasing implies standard procedures (e.g., BLAST, InterPro, Pfam), but it does not spell out which databases or cutoffs were used. We recommend specifying the thresholds for “standard annotation” (E-values, coverage) and to provide a brief rationale for how uncharacterized or borderline hits were handled.

Gene Organization in Subfamilies (Figure 1)

The location, tandem duplication, and/or scattering of multicopy genes is not systematically described. We suggest to indicate whether the copies in each subfamily cluster together in the genome (suggesting recent duplications) or are interspersed (suggesting older acquisitions). Consider dividing large subfamilies into sub-clusters (“sub-subfamilies”) to track possible separate origins.

Code availability and documentation:

there is currently no info on the used code/analysis code. This is essential, needs to be documented, submitted for review and ideally deposited on Github etc.

Minor Points

Figure S5. It is unclear whether this is a predicted or experimental structure. Please state either the PDB or how it was predicted along with the confidence scores.

Figure S7. Please provide the ipTM value(s) and PAE plot for the multimer.

Figure S9. The caption says “1053 and 1054: pTM = 0.5, pTM = 0.61 and 1055 and 1056: ipTM = 0.58, pTM = 0.63”. Should it say “ipTM = 0.5” in the first pair like the second pair?

Reviewer #2

(Remarks to the Author)

I co-reviewed this manuscript with one of the reviewers who provided the listed reports. This is part of the Nature Communications initiative to facilitate training in peer review and to provide appropriate recognition for Early Career

Researchers who co-review manuscripts.

Reviewer #3

(Remarks to the Author)

As the title “Convergent evolution of viral-like Borg archaeal extrachromosomal elements and giant eukaryotic viruses” conveys, this manuscript is a comparison of features in between Borg elements and giant viruses in the phylum Nucleocytoviricota. The manuscript also provides arguments as to why Borg elements may be viruses rather than plasmid genomes.

The methods used included using previously identified protein families for further functional inference, protein structure prediction, phylogenetic analysis and transcripts from soil samples.

There were no labeled sections for the Introduction, Results and Discussion. However, the two paragraphs following the MAIN text header served as the introduction summarizing knowledge about the Borg elements. Given the title, some introduction of the giant viruses is necessary. Particularly important for this manuscript is that large genome size (e.g. greater than 1 Mb) has arisen 3 separate times within the Nucleocytoviricota (Koonin EV, Yutin N. 2018; Koonin EV, Yutin N. 2019) and the diversity in cell shape, structure and diversity between this giant virus groups. The authors lump these 3 distinct lineages into one category. Are the Borgs more similar to one of these types of giant viruses than the other? There are also many papers noting similarities between the giant viruses and jumbo phages. In the analysis it appears there are more similarities between bacteriophages than with the 3 giant virus lineages.

The 65 NCLVs included appear to be from Table S11. A quick scan indicates that most genomes are from the Imitervirales. Only 1 genome is from the 2nd giant virus lineage associated with the Pimascovirales and none are from the 3rd lineage comprising the Pandoravirales. Thus this analysis is really a comparison of giant viruses in the Imitervirales with Borgs rather than a comprehensive comparison to NCLV giant viruses. This leads to questions regarding the comparisons to “giant viruses” throughout the text.

Overall, the manuscript would be clearer if the results and discussion were separated. There are some nice insights into the Borg protein functions. In the discussion, compare Borgs to other large elements, plasmids and viruses.

Although the data is enticing, Borgs have not yet been shown to be viruses. Thus the sentence in the Abstract “If Borgs are giant archaeal viruses they would fill the gap in the tri(um)virate of giant viruses of all three domains of life.” is premature, but would be appropriate conjecture in the discussion following a comparison of Borgs with the largest eukaryotic and bacterial viruses.

Figure 1. Legend Y-axis should read Number of proteins per Borg)

Figure 6. The Nucleocytoviricota included graphs are all in the order Imitervirales. This is just one of the 3 lineages of giant viruses as noted above.

Table 2.

This again suffers from 64 of the 65 genomes coming from the Imitervirales.

Reviewer #4

(Remarks to the Author)

Authors explored the functional properties of a lineage of very large extrachromosomal and linear double-stranded DNA elements (Borgs) associated with an archaeal genus (*Methanoperedens*) lacking culture representatives. A main objective was to determine if these intriguing elements, which have a large majority of genes lacking any functional annotation, correspond to mega plasmids or giant viruses. If the later, then it would be the first documented occurrence of such viruses with very large genomes in the archaeal domain. Authors predicted protein 3D structures to expand the functional analysis of multi-copy genes (~10k proteins) among 17 high-quality genomes, described functional families widespread in these genomes, and finally compared genomic features of Borgs and giant eukaryotic viruses. They concluded that “Borgs have numerous features suggestive of lifestyles somewhat analogous to those of giant eukaryotic viruses and we conclude that the weight of evidence now supports a more viral like than plasmid-like lifestyle”.

In the first part of results, authors described important functional families widespread in the genomes of Borgs, which provide insights into the functional capability of those elements. Widespread functional families include deSAMPylases and glycoconjugates, which the authors argue might be relevant to a viral lifestyle. Most importantly, authors further explored multi-copy genes annotated as possible viral capsid proteins, already identified by Schoelmerich et al. (2024). The most interesting functional hit is for single-jelly-roll proteins of an archaeal DNA virus named *Haloarcula californiae* HCIV-1. The spread of genes corresponding to single jellyroll like proteins, their high expression levels, and predicted hexamer assemblies into a capsid-like sheet are certainly interesting and relevant to our understanding of Borgs.

In the second part of results, authors compared genomic features of Borgs and giant eukaryotic viruses of the phylum Nucleocytoviricota. They suggest that the common features they identified here and in previous publications (megabase-scale linear genomes with inverted terminal repeats, genomic repertoires for energy metabolism, central carbon compound

transformations and translation, and finally pervasive direct repeat regions) correspond to convergent evolution.

I have two concerns related to the structure and conclusions of the manuscript, which are described below. Most importantly, authors did not provide convincing evidence (hallmark genes of viruses absent in cellular genomes) that Borgs are viruses.

(1) Lack of signal for hallmark genes of the virion morphogenetic module and limited prominence of single-jelly-roll proteins:

Authors did not detect hallmark genes of viruses that are absent in cellular genomes. Critically, single jelly-roll proteins are widespread in cellular genomes (<https://www.pnas.org/doi/full/10.1073/pnas.1621061114>), and only correspond to the major capsid proteins (MCPs) for known viruses with very small genomes. In viruses with large genomes, the single jelly-roll proteins are cementing the particle (decoration proteins) but do not represent its main component. There are three remaining possibilities: (1) Borgs are the first known viruses with large genomes that only have single jelly-roll proteins as main particle components (single jelly-roll proteins would be the MCPs), (2) their MCPs have not been identified thus far, (3) or they are not viruses (no particle formation). Because single jelly-roll proteins can have cellular functions, no convincing evidence is provided that would demonstrate that Borgs can form particles.

(2) Lack of comparisons between Borgs and plasmids

Authors provided interesting genomic feature comparisons between Borgs and giant eukaryotic viruses. Yet, no similar comparisons were provided to compare Borgs and plasmids. Since none of the shared features between Borgs and plasmids are described, it is difficult to assess the relevance of a main conclusion of the study: “we conclude that the weight of evidence now supports a more viral like than plasmid-like lifestyle”.

Authors may address my concerns by adding perspective on genomic feature similarities and differences between Borgs and plasmids, and by toning down the importance of capsid-like particles detected in Borgs, as those can also be present in cellular genomes for other functions. Of course, these points conflict with the main claim of the study, and the authors may have to adjust the main objectives of the manuscript, which aside from the viral angle certainly provides new insights into the functional capabilities of very intriguing extrachromosomal archaeal DNA elements. Those may or may not be giant viruses, and there are limits to what bioinformatics can deliver. It would be astounding if someone was to observe viral particles with microscopy, thus demonstrating that Borgs are indeed giant archaeal viruses.

Version 1:

Reviewer comments:

Reviewer #1

(Remarks to the Author)

I thank the authors for the substantial and constructive revision. I have only a small set of outstanding issues, all focused on data clarity.

-I could not locate the new Supplementary Tables

Figure 5: The legend lists ipTM values for the capsid pentamers and hexamers. Is this the global ipTM or is the range given for all pair-wise ipTM for all models? Including pair-wise ipTM values is instrumental. Could you include the pairwise values for at least 1 example? Perhaps as a heatmap. Moreover, at least the most important PAE plots should be included in the figure to illustrate the modelling quality.

Figure S3: As above, specify whether the quoted ipTM is global or interface-specific. Because Zn was incorporated in these models, confirm that exactly one Zn²⁺ per protomer was modelled and note this in the caption. If alternative stoichiometries were explored, indicate which configuration produced the reported scores.

In the Supplementary Data file, line 31 reads “ipMT = 0.94”; this should be “ipTM = 0.94”.

Reviewer #2

(Remarks to the Author)

Reviewer #3

(Remarks to the Author)

The authors made a number of changes in response to the reviewer comments that improved the manuscript. Most reviewer comments were satisfactorily addressed and the remainder I am willing to go with the author perspective.

Reviewer #4

(Remarks to the Author)

I thank the authors for summarizing some of the main differences between BORGs and plasmids. Since this information was relevant to me, I wonder if the authors could very briefly state in the introduction that major differences are observed between BORGs and plasmids, just before pivoting to discussing features of large eukaryotic viruses. I realize the manuscript is very long, and I only suggest this for the sake of clarity among future readers of the publication.

Overall, I commend the authors for dedicated so many efforts in trying to clarify the nature of those genomic elements using bioinformatics. Despite a lack of signal for some of the most critical viral hallmark genes, the manuscript is full of relevant insights regarding the genomic makeup of BORGs and comparison with evolutionary unrelated viral genomes is both original and relevant.

Finally, I wish the authors best of luck in their efforts to detect particles related to BORGs despite the difficulty of this endeavour.

Tom Delmont

Version 2:

Reviewer comments:

Reviewer #1

(Remarks to the Author)

I thank the authors for their excellent work. I do not have any further queries.

Reviewer #2

(Remarks to the Author)

**REVIEWER COMMENTS**

*Reviewer #1 (Remarks to the Author):*

*This manuscript analyzes the proteomes of giant extrachromosomal elements—Borgs—using*
*large- scale in silico protein structure prediction (AlphaFold2/3). By clustering thousands of Borg*
*proteins into subfamilies, the authors identify repeated functional themes, including metabolic*
*enzymes, deubiquitinase- like (deSAMPylase) metalloproteases, and capsid- like proteins. The*
*study builds a case that Borgs may share features with large dsDNA viruses (Nucleocytoviricota-*
*like), prompting discussion of whether Borgs should be classified as giant archaeal viruses or*
*remain viewed as plasmid- like elements.*

*Overall, the manuscript provides a valuable resource by applying large- scale AlphaFold*
*predictions to an extraordinary set of archaeal extrachromosomal elements.*

*We thank the reviewer for their positive comments, appreciation of the scale of the data analysis,*
*careful review of our manuscript, and in-depth questions about thresholds for structure*
*predictions.*

*Our central criticisms revolve around clarifying how confident the authors can be about predictions*
*by establishing thresholds for the used scores, specific multimeric states (especially putative*
*capsid proteins), and deeper mechanistic/structural details for ambiguous cases (e.g.,*
*metalloproteases). Addressing these points should significantly strengthen the manuscript.*

*We concur that our approach was insufficiently explained, and agree that adding the requested*
*information has strengthened the manuscript. The study was predicated on the ideas that (i) the*
*presence of proteins in multicopy in Borg proteomes would be an indicator of importance, and (ii)*
*that *in silico* structure prediction could inform our understanding of likely protein functions. Each*
*of the groups of multicopy proteins with the highest prevalence was examined in detail, starting*
*with direct visualization of the structures in ChimeraX. Confidence values were directly visualized*
*across each structure and evaluated in the context of alignments, sequence conservation and*
*presence of active site residues. Additional analyses are mentioned below and in the revised*
*methods.*

*A thresholding approach may have only focused on confident predictions (pLDDT > 0.7).*
*However, pLDDT scores are calculated from values for all residues in each protein and so model-*
*wide values reflect scores for both ordered and disordered regions, such as local regions of*
*intrinsic disorder, which are a well documented feature of many Borg proteins; Schoelmerich et*
*al. (REF). We use median rather than mean pLDDT values to limit this effect.*

*We provide all structures and information about them for future use by the research community,*
*including extensive tabulations of statistics. We now report that confident predictions (median*

pLDDT > 0.7) were generated for 69% of proteins. Of these, 36% had potentially informative hits
in PDB (based on bitscores of ≥ 200) with an average pLDDT of 93.92.

As noted above and referred to below, we have added considerable information and explain our
approach in a much more comprehensive way in the Methods, as follows:

“Protein structure prediction and functional analysis

The genomes of Orange, Black, Green, Amber, Amethyst, Cobalt and Ruby Borgs were selected
as representative of Borg diversity based on their relatedness, as inferred from the phylogeny
based on 40 single-copy universal genes that we reported previously (Schoelmerich et al. 2024).
For each protein in these seven Borg genomes, ColabFold (--amber --templates --num-recycle 3)
(Mirdita et al. 2022) run locally was used to predict AlphaFold2 structures that were evaluated
initially for similarity to proteins in the Protein Database (PDB, <https://www.rcsb.org/>). In total, we
folded 8847 Borg proteins as monomers. The mean and median pLDDT values for all reported
structures and other statistics are provided in Supplementary Table 1 and the median pLDDT is
reported protein by protein for each of the seven Borg genomes. Information is also found in each
.pdb file and data folder for each protein (Supplementary Information). We prioritized analysis of
structures for proteins from subfamilies that occurred multiple times in the Borg genomes
(multicopy subfamilies) and selected the most prevalent multicopy subfamilies for detailed
analysis. Phylogenetic analyses (see below) were performed for the largest subfamilies and
related subfamilies to distinguish clades and group subfamilies with essentially the same
structures and inferred functions.

To obtain potential functional insights for proteins from multicopy subfamilies, we identified the
structure model with the highest score (model wide pLDDT) to search against the Protein Data
Bank (PDB) for structural similarity using Foldseek (<https://search.foldseek.com/search>). Most
models selected for further consideration had a median model wide pLDDT of ~70, a value that
meets the accepted definition of a high quality structure (AF2). However, we did not limit our
analyses to these models as median scores can be strongly influenced by local disordered
regions (e.g., intrinsically disordered regions that are prevalent in Borg protein (Schoelmerich et
al. PLoS Biology, REF), yet other parts of the structures can provide functional clues. We chose
to use median rather than mean value of pLDDT to reduce this effect. We favored a more inclusive
approach, given the likely novelty and wide divergence of Borg proteins compared to
experimentally studied proteins.

To robustly assess proteins with potentially credible hits in the PDB, we visualized and manually
investigated structural predictions for proteins representative of each protein subfamily discussed
in this manuscript using UCSF ChimeraX (version 1.8, <https://www.rbvi.ucsf.edu/chimerax>).
Although we prioritized structural predictions with bit score values to a PDB structure of >200,
subfamilies with no relatively high scoring matches to a PDB entry were also evaluated, so long
as the bitscores were > ~50. These (and some other) models were searched against the
AlphaFold database that includes predicted as well as experimentally validated structures using
Foldseek. Analyses included information such as the local per-residue confidence (pLDDT) and

agreement between local and global structural features of the PDB hit(s), as well as the
distribution of charged and hydrophobic residues and amino acids of interest (e.g., associated
with active sites). As we focused on multi-copy proteins, visualization also included residue by
residue sequence conservation that was rendered from protein sequence alignments. In cases
where portions of the protein were apparently rotated relative to the PDB reference, that portion
was extracted and realigned. Unexpected regions (e.g., extra domains) were extracted and the
structures compared to existing domains (in cases where domains were duplicated) and to public
data using Foldseek.

For monomers, we investigated convergence of the five models (rank 0 - rank 4) by comparing
the pLDDT values for the models generated using Colabfold. We concluded that AlphaFold
reached convergence if all five models converge on the same structural solution. To evaluate this,
we used TM-align (REF) for all pairwise comparisons (335,911 comparisons). When requiring
100% alignment over the entire length, 91,677 were essentially identical (TM score 0.5 - 1 is
considered essentially the same score). 183,377 proteins had 80% alignment and 80% identity.

For multimer predictions, we used AlphaFold3 (Abramson et al. 2024) with default settings and
assessed the pLDDT, PTM (overall model reliability, derived from the predicted alignment error,
PAE) and iPTM (interface pTM, the accuracy of the predicted relative positions of the subunits
forming the protein-protein complex). For analysis of capsid-like proteins, multimers of 2 - 9
subunits were evaluated based on iPTM scores and the highest scoring multimer was chosen as
the most likely state. The iPTM scores >0.8 were considered confident, following AlphaFold
guidelines. We analyzed the highest confidence prediction for each model using ChimeraX to
visualize the local per-residue confidence (pLDDT), sequence conservation based on sequence
alignments, and agreement between local and global structural features of the PDB hit. We
assessed convergence of the five models, requiring similar high (> ~80) iPTM scores for all five
of the models generated after three recycles. Buried areas between multimer domains were
estimated as a measure of interaction confidence using Chimera X. For putative deSAMPylases,
complexes with SAMP were predicted from representatives of the major clades defined by
phylogenetic analysis of protein sequences.

To test whether multimer prediction confidence could be improved for a putative deSAMPYlase
complexed to SAMP, we augmented the ColabFold AlphaFold3 prediction using an MSA that
included all Borg sequences and sequences from Methanoperedens and other archaea and an
MSA for credible SAMP proteins from similar sources (Supplementary Data). Calculations were
performed with up to 100 recycles. As the difference between 20 and 100 recycles was nominal
and no improvement was achieved, we did not perform an analogous analyses for all of the
SAMP-deSAMPylase multimers.”

Interestingly, of the 8847 models generated, 6,073 (63%) had predicted structures with pLDDT
scores of >70. Of these confident structures, 1490 had bit scores of <50 (25%), and 2936 (48%)
had bit scores <100. This information has been added to the manuscript. This likely reflects the
lack of representation (or substantial divergence) of many Borg protein structures compared to
those in the PDB, and motivated careful consideration of proteins with relatively low bitscores.

Bit scores of <100 proved informative for some subfamilies, especially putative de-SAMPylase
metalloproteases, for which many confident structure predictions were generated. To support their
identification, structural comparisons, active site analyses and multimer predictions with SAMP
were performed. As noted above, we used the per residue pLDDT confidence metric to guide our
interpretations of the confidence in the active site region. As requested below (and noted above),
median pLDDT values for each protein are listed sequentially in the genome files (Supplementary
Tables 2-8).

The question related to ‘*specific multimeric states (especially putative capsid proteins),*’ is
responded to below.

*Major Points*

*Thresholds for prediction scores*

*Monomers:*

*The manuscript cites pLDDT >70 as “confident” but does not clearly indicate whether this refers*
*to a global (average) value or per- residue cutoffs. Likewise, pTM and ipTM thresholds are*
*presented as indicative of confidence, but specific threshold validation is missing. The authors*
*should clarify how per- residue pLDDT scores were aggregated to define a protein as “confident,”*
*whether local regions must meet the same threshold, and how this was weighted against the pTM*
*score. A clear thresholding pipeline needs to be defined here, and statistics of how many proteins*
*passed thresholds are needed per software used (see below).*

Unlike the pTM aggregated score per protein, pLDDT is a per residue measure. We used per
residue information for visualized proteins and median pLDDT scores as measures of fold
confidence (i.e., pLDDT values for each protein were aggregated by calculating the median value
for all protein residues). In general, we considered median pLDDT values >0.70 as confident,
PTM >0.6 as confident, and ipTMscores >0.8 as confident. We provide residue by residue pLDDT
values and median values (etc) in the supplementary files.

The predicted structures were not subsampled into “confident” and “not-confident” categories prior
to detailed analyses because phenomena such as sequence/ structure/local disorder likely make
no single aggregated value reliable for any protein structure prediction. We performed manual
analyses that included (and excluded) consideration of subregions of proteins with lower median
scores. This is now indicated, and the rationale appears in text.

In the revised version, multimers are only predicted using AlphaFold 3 (this was also used for
targeted reexamination of protein monomers of interest). We have clarified the methods, as
above. Please also see responses below.

*Multimers:*

*The authors need to verify their ipTM thresholds, especially on archaeal protein datasets,*

For multimers, we added details to the Methods, included above.

For putative capsid proteins, the n-mer representing the most likely multimerization state (or
states) had the highest ipTM and pTM scores. We only present detailed cases where all five
models had ipTM values of $> \sim 0.8$ (now listed in **Table S12**). We include the ipTM, pTM and for
the highest scoring multimer, the range of RMSD values for models 0-4 (pruned atom pairs and
all) in this new Supplementary Table. Additional information (including estimated buried area)
has been added to the caption for **Figure 5**.

Also to address convergence, we tested whether the 5 calculations for each sequence generated
essentially the same structure using Foldseek similarity. To illustrate: for Amethyst_1056 homo-
hexamer (not the highest scoring capsid-like protein), rank 0 - rank 4 models were superimposed
as shown below; the PAE plot is also provided. These are not included in the manuscript, but
added here to address reviewer concerns. The per residue pLDDT for this protein can be viewed
in the new version of Figure 5.

We have now more thoroughly investigated the confidence of the SAMP - E1 and SAMP -
deSAMPylase multimer predictions and the results are interesting. The *Methanoperedens* SAMP
- E1 and SAMP - deSAMPylase multimers are very confidently predicted and values are now
provided (**Table S10**). Across the tree of Borg deSAMPylases, the multimer structure prediction
confidence values are quite variable and some rate as not confident, despite the protein displaying
the expected active site residues, confident Zn binding to the conserved active site, and
localization of the diglycine SAMP tail at the active site. Some Borg SAMP - deSAMPylase
multimers are very confidently predicted and we feature examples in a new version of **Figure 2**.
Confidence prediction does not seem to vary with phylogenetic placement. Results for a subset
of possible deSAMPylases are now reported in supplementary **Table S10**. We conclude that
protein duplication followed by divergence may lead to differences in function (despite retaining
active site residues and deSAMPylase topology) or reflect some unrepresented phenomenon
such as post translational modification. We have revised the text to reflect these findings.

Also relevant to the next comment: For one case of a non-confident multimer prediction generated
by AlphaFold3, we tested whether confidence could be improved by augmentation of the MSA
used in ColabFold. To do this, we generated a MSA for credible SAMP proteins and used it,
along with the MSA for the putative Borg and archaeal deSAMPylases in calculations with up to

100 recycles (the difference between 20 and 100 recycles was nominal). No improvement was
achieved.

*Model convergence*
*The authors specify that a limited number of three recycles are run in ColabFold. While sufficient*
*for easy-to-predict folds, especially complex or multi- domain proteins, 3 recycles may or may not*
*be sufficient for models to converge. This can be typically seen by not converging pLDDT plots*
*for the five models. The authors need to define thresholds for model convergence and what they*
*did with predictions that were not sufficient (e.g. running more recycles).*

We agree with the reviewer that model convergence is an important consideration for confident
assessment of structure predictions. As Colabfold seeds independently 5 jobs that run to
convergence (after 3 recycles), we considered that Colabfold “succeeded” if all 5 structures have
similar high pLDDT values.

In response to reviewer comments, we have now evaluated convergence of the 5 models for all
monomer structures generated using Colabfold after 3 recycles. This was performed pairwise (all
combinations) for all 5 models using TM-align within Foldseek. Over 335,911 comparisons and
requiring 100% alignment over the entire length, 91,677 were effectively identical, 99,017 were
$\geq 95\%$ identical, 106,097 were $\geq 90\%$ identical, and 113,119 were $\geq 80\%$ identical. 183,377 proteins
had 80% alignment and 80% identity. Clearly, some models did not converge. This might be due
to the presence of low complexity or intrinsic disordered regions in which some subset of the
predictions were seeded. The conclusions we present (e.g., putative deSAMPylases, DPMS,
histones) are for proteins for which reasonable to good structural matches were identified,
supported via a variety of additional analyses, as described above. The analysis is now provided
in the text.

*Additionally, browsing through the pLDDT plots of the predictions, we have a hard time following*
*the argument that about 60% of predictions were high quality by evaluating model convergence*
*and overall pLDDT distribution.*

The statement that $>60\%$ of the predictions are of high quality is based on the median pLDDT
value, of ≥ 70 . Actually 68.6% of the generated structures met this threshold. Please see **Table**
**S1** and the plots in **Figure S1**.

*Use of AF vs. Colabfold*
*The authors used a mixture of prediction software. Scores between models cannot directly be*
*compared, and they perform differently, especially in hard-to-predict cases where MSAs are thin.*
*The authors need to state a clear strategy for which protein and which software was used. E.g.,*
*using vanilla AlphaFold 2 when Colabfold gave low-confidence predictions due to sparser MSA.*

All monomer protein structures were predicted using Colabfold (vanilla AlphaFold 2 was not
used).

*Moreover, large parts of the Methods section on structure prediction are ambiguous. The authors*
*again need to develop a clear strategy with thresholds and a data management solution to share*
*the data with all required metadata.*

Regarding methods, please see responses above and below. In terms of data management, we
now provide files that contain data for every protein. The folders (one per protein) are bundled in
a Zenodo archive (DOI <https://zenodo.org/records/15795806>). The folders for multimers
discussed in the manuscript are provided in **Supporting Data 4**.

*Structural similarity search*
*It is unclear why only searches against the pdb were performed for most predictions. This should*
*be expanded to the full AFDB as well as against the BFVD from the Steinegger lab*
*(<https://bfvd.foldseek.com/>) to illuminate viral similarity.*

We focused first on PDB matches because the PDB is a collection of experimentally validated
structures. However, in depth investigations for specific protein multicopy subfamilies also used
comparisons to the full AFDB. This has been clarified in the methods.

We had previously tested the utility of BFVD for identification of virus-like proteins. We have
redone this analysis for all Borgs as requested, and find very low similarity to any structures in
this database: the highest bitscore is 67 (an endopeptidase related to proteins from
*Methanoperedens*) and only 13 had bitscores > 50. The lack of similarity to proteins in this
database is perhaps unsurprising, given the novelty of these archaeal extrachromosomal
elements. This information has been added to the text.

*Accessibility of predictions*
*All prediction data needs to be available. Currently, only pdbs and pLDDT files are shared but*
*PAE plots and MSAs are missing, as well as all json output files. This is absolutely crucial to*
*evaluate prediction quality and to figure out what settings were used. **We suggest organizing***
***them in folders per protein** and not folders for each data type as done now.*

We now provide the full data sets (including PAE plots and json output files) for all proteins (see
above), and the MSAs for proteins featured in this study (e.g., SAMPs, de-SAMPs).

*Bitscore Cutoffs (>200) and E-Value Thresholds in FoldSeek/Dali*
*The authors occasionally accept matches in the 50–200 bitscore range when no better hits were*
*found. While that can be reasonable for a domain of unknown function, it is inherently more error-*
*prone. For these borderline scores, it would be prudent to confirm via domain- by- domain*
*alignment or partial alignments (e.g., discarding disordered N/C- terminal segments) to ensure*
*the match is *truly* to the same fold.*

No functional predictions with scores in this range were simply accepted. In every case, the
models were evaluated using ChimeraX, taking into consideration the aligned and non-aligned
portions, extracting extra domains, refolding segments etc. In addition, active site residues were
analyzed for proteins in the main categories discussed. This is described in the (revised) methods.

*A more explicit mention of “secondary checks” for borderline structural similarities would help*
*readers judge the reliability of each functional claim.*

We outline a variety of additional sources of information used; please see the revised Methods.

*Confidence in Hexameric Capsid Proteins (Figures 5C, S9)*

*The manuscript relies on a single global ipTM (or pTM) to claim that its predicted capsid proteins*
*form hexameric assemblies. However, simply reporting one global metric does not confirm a*
*stable interface for every subunit pair, nor does it clarify whether other lower- order oligomers*
*(e.g., dimer, trimer, pentamer) might also form. In many icosahedral viruses, pentameric as well*
*as hexameric capsomers are required, and the lack of high- confidence pentamer predictions*
*may cast doubt on the capsid hypothesis.*

This was an excellent suggestion! Initially, we calculated a variety of oligomeric states for a couple
of putative capsid proteins and found very substantially better results for homohexamers and
were reassured when high scores were obtained for other proteins as hexamers. However, upon
doing calculations for homo -dimers, -trimers, -tetramers, -pentamers, -hexamers, -7mers, -
8mers, 9-mers brought light that some indeed have better scores for homo-pentamers than homo-
hexamers. Interestingly, some proteins may be capable of forming stable homopentamers (e.g.,
Amethyst 1054 best model homopentamer ipTM 0.95 and homohexamer ipTM 0.91). This is a
feature of many icosahedral virus capsid proteins. Other icosahedral viruses (e.g., adenovirus)
use two different proteins for the pentameric and hexameric capsomers, which is another
possibility, given the multiplicity of capsid-like proteins. This information has been added to the
manuscript.

**Table S12** now provides the multimer statistics for Black and Amethyst Borg capsid-like proteins
as well as some combinations of pentamers and hexamers etc. For each calculation, the table
reports ipTM, pTM, structural configuration predicted, model convergence RMSD models 0-4 and
for the highest scoring calculation, the buried area between subunits.

*We strongly suggest providing pairwise ipTM Data and full PAE plots for all assemblies and*
*illustrating local interface predictions to confirm a single stable complex rather than partial*
*interactions for all shown multimer predictions (not just potential capsid proteins) in the*
*manuscript.*

To address this comment, we have included the full prediction dataset for each multimer
calculation for the capsid-like and putative deSAMPylase proteins. These files provide all of the
requested ipTM and PAE data. For putative capsid-like proteins we provide evidence of sequence
conservation of residues in capsid multimer interfaces (**Figure 5D**), high model confidence scores

and model convergence. Providing molecular illustrations for all multimers is beyond the scope of
this paper, but we note that there are cavities lined with hydrophobic residues on both sides in
the Amethyst Borg pentamer, suggesting something else binds there (e.g., lipids or another
protein such as a fiber). This remains too speculative to be added to this manuscript.

The findings of the reanalysis of the de-SAMPs and new reporting is summarized above.

*This will allow readers to evaluate whether the predicted subunit–subunit contacts are uniformly*
*well- resolved rather than relying only on a single ipTM for the entire complex. We also strongly*
*suggest attempting predictions for dimer, trimer, pentamer, and hexamer assemblies.*

As above, these calculations were performed and the results are tabulated and reported.

*If smaller (or pentameric) stoichiometries show very low confidence, the classification of these*
*proteins as bona fide icosahedral capsid proteins remains uncertain. In other words, the lack of*
*stable pentameric assemblies in particular may suggest an alternative biological function or a less*
*conventional capsid organization.*

We are aware of the expectations regarding capsid oligomer mixtures in some (e.g., eukaryotic)
icosahedral viruses, but note that these cannot define expectations for novel archaeal viruses. In
fact, archaeal viruses are known to have unique architectures (e.g., spindle shaped viruses:
<https://doi.org/10.1016/j.cell.2022.02.019>). Nonetheless, as noted above, we now report support
for pentameric as well as hexameric assemblies.

Regarding smaller stoichiometries, many of the multimers <5 are clearly parts of larger
assemblies (e.g., a trimer that is half of a hexamer, see **Table S12**).

*Potential metalloproteases (Figure S3)*

*The study references Zn- metalloprotease functionality but does not explicitly show predictions*
*with Zn. The authors should explicitly run AlphaFold3 (or equivalent pipeline) with Zn to test*
*whether the predicted active site geometry is consistent with Zn binding. Also the catalytic triad*
*or metal- binding site must have high local pLDDT consistently, or else structural claims about*
*catalytic residues (also in the putative deSAMPylases) may be on shaky ground.*

We now include Zn in the calculations and find that its binding is very well supported (with high
prediction confidence as well as sequence conservation). This information has been added to the
caption of **Figure 2** (e.g., Orange 866: ipTM: 0.97, PTM: 0.89). Zn binding to proteins with active
site variants is also supported, and confidence values have been added to the **Figure S3** caption.
Remade figures show Zn in the active site (and in proximity to the SAMP diglycine tail).

*Standard Annotation Methods*

*The manuscript's phrasing implies standard procedures (e.g., BLAST, InterPro, Pfam), but it does*
*not spell out which databases or cutoffs were used. We recommend specifying the thresholds for*

“standard annotation” (E- values, coverage) and to provide a brief rationale for how
uncharacterized or borderline hits were handled.

These annotations are from a prior publication for which the SOM presents the annotations. For
details, see Schoelmerich et al. (2024). These data are provided but they were not featured in the
current analysis.

*Gene Organization in Subfamilies (Figure 1)*

*The location, tandem duplication, and/or scattering of multicopy genes is not systematically*
*described. We suggest to indicate whether the copies in each subfamily cluster together in the*
*genome (suggesting recent duplications) or are interspersed (suggesting older acquisitions).*

We have added the subfamily assignment for every protein in the spreadsheets for the 7 Borg
genomes (**Tables S2-S8**) and identify instances of adjacent genes that encode proteins assigned
to the same subfamilies (new **Tables S19, S20**). The sequentially encoded subfamilies are
summarized and we provide possible functional annotations based primarily on structure analysis.
In most instances there are only two sequential proteins (sometimes three), and the same
subfamily often occurs sequentially in multiple Borg genomes. For highly replicated (muticopy)
subfamilies, the genes are generally distributed across genomes.

*Consider dividing large subfamilies into sub- clusters (“sub- subfamilies”) to track possible*
*separate origins.*

The manuscript includes trees for the major subfamilies and shows the breakdown into clades,
possibly indicative of separate origins. As noted, the clades do not simply align with subfamilies,
we have not pursued sub-subfamilies.

*Code availability and documentation:*

*there is currently no info on the used code/analysis code. This is essential, needs to be*
*documented, submitted for review and ideally deposited on Github etc.*

The only new code is an update of the tandem repeat finder. As mentioned, this is already
available on Github. No new code was generated to make any display or analysis figures.

*Minor Points*

*Figure S5. It is unclear whether this is a predicted or experimental structure. Please state either*
*the PDB or how it was predicted along with the confidence scores.*

The figure reports features of structural predictions for DPMS proteins. This is now clarified in
the caption.

*Figure S7. Please provide the ipTM value(s) and PAE plot for the multimer.*

Figure S7 is a diagram of a split gene so I think the reviewer refers to Figure S6. The caption has
been amended with ipTM, pTM, as well as estimation of the buried area between subunits.

Figure S9. The caption says “1053 and 1054: pTM = 0.5, pTM = 0.61 and 1055 and 1056: ipTM
= 0.58, pTM = 0.63”. Should it say “ipTM = 0.5” in the first pair like the second pair?

The missing i is now added, thank you

*Reviewer #2 (Remarks to the Author):*

*I co-reviewed this manuscript with one of the reviewers who provided the listed reports. This is*
*part of the Nature Communications initiative to facilitate training in peer review and to provide*
*appropriate recognition for Early Career Researchers who co-review manuscripts.*

This is a great approach; your mentee did an incredible job with this review.

**Reviewer #3 (Remarks to the Author):**

*As the title “Convergent evolution of viral-like Borg archaeal extrachromosomal elements and*
*giant eukaryotic viruses” conveys, this manuscript is a comparison of features in between Borg*
*elements and giant viruses in the phylum Nucleocytoviricota. The manuscript also provides*
*arguments as to why Borg elements may be viruses rather than plasmid genomes.*

*The methods used included using previously identified protein families for further functional*
*inference, protein structure prediction, phylogenetic analysis and transcripts from soil samples.*

*There were no labeled sections for the Introduction, Results and Discussion. However, the two*
*paragraphs following the MAIN text header served as the introduction summarizing knowledge*
*about the Borg elements.*

“Introduction” and “Results” subheadings have been added. Other sections already have
subheadings.

*Given the title, some introduction of the giant viruses is necessary.*

We now briefly introduce giant viruses in the Introduction, in part by moving text up from a later
section.

*Particularly important for this manuscript is that large genome size (e.g. greater than 1 Mb) **has***
***arisen 3 separate times within the Nucleocytoviricota (Koonin EV, Yutin N. 2018; Koonin***
***EV, Yutin N. 2019) and the diversity in cell shape, structure and diversity between this giant virus***
*groups. The authors lump these 3 distinct lineages into one category. Are the Borgs more similar*
*to one of these types of giant viruses than the other?*

We have discussed features that occur in both Borgs and members of the *Nucleocytoviricota*, and
whether Borgs are more similar to any particular lineage depends on what features are
considered. For example, genes involved in metabolism are most prevalent in the *Imitervirales*
(Aylward et al., PLOS Biology, 2021), and so this group bears the most similarity to Borgs in terms
of the presence of central carbon metabolic genes. Tandem repeats are broadly distributed across
nucleocytovirus lineages (see below). Traits found in this lineage occur across a continuum; for
example, the prasinoviruses have relatively small genomes (~200 kbp) but still encode several
genes involved in central carbon metabolism. We therefore believe it makes the most sense to
discuss comparisons of Borgs with nucleocytoviruses as a whole rather than specifically discuss
only groups with larger genomes.

*There are also many papers noting similarities between the giant viruses and jumbo phages. In*
*the analysis it appears there are more similarities between bacteriophages than with the 3 giant*
*virus lineages.*

We do not agree that Borgs share more similarities with jumbo/giant phages than with giant
viruses. As we note in Al-Shayeb et al. Nature, giant bacteriophages encode organismal genes,
but (beyond those involved in replication and nucleotide metabolism), most prevalent are
translation-related. In some cases, huge phage genomes have far more non-tRNA translation
genes than found in any Borg. However, what really sets Borgs apart is the incredible repertoire
of metabolic genes that are not present in huge phages. Among many other distinctions is the
prevalence of tandem direct repeats in the linear genomes of Borgs and many Eukaryotic giant
viruses (but not phages). However, we concur that there are interesting features shared by some
giant phages and giant eukaryotic viruses, including viral factories and this is mentioned in the
text.

Mention of Borg and eukaryotic ubiquitin-related functions was accidentally omitted from the
manuscript; this has now been included.

*The 65 NCLVs included appear to be from Table S11. A quick scan indicates that most genomes*
*are from the Imitervirales. Only 1 genome is from the 2nd giant virus lineage associated with the*
*Pimascovirales and none are from the 3rd lineage comprising the Pandoravirales. Thus this*
*analysis is really a comparison of giant viruses in the Imitervirales with Borgs rather than a*
*comprehensive comparison to NCLV giant viruses. This leads to questions regarding the*
*comparisons to “giant viruses” throughout the text.*

Confusing perhaps is that this Table listed the NCLVs used in tandem repeat analyses. Overall
comparisons to NCLV gene content involve the full diversity of NCLV types (although metabolic
genes are generally more prevalent in Imitervirales, which includes most of the well known giant
eukaryotic viruses - as discussed above).

Regarding the genomes used for the tandem repeat analyses: we concur that some groups were
not represented. The original analysis was restricted to a set of 65 genomes of > 500 kbp in
length, so broadly comparable with Borgs. We have added a new analysis that documents repeat
statistics for a much larger and more representative set of genomes. **Table 2** has been augmented

with statistics for a select diverse set of NCLVs > 500 kb in length and for diverse NCLV genomes
> 10 kbp in length. Repeat analysis datasets are now presented in **Table S15 - S18**). After
accounting for genome fragments, the composition of the new dataset is: 27 Algavirales, 25
Asfuvirales, 9 Chitovirales, 141 Imitervirales, 20 Pandoravirales, 84 Pimascovirales, 2
incertae_sedis.

Order classifications have been added to the sequence list in **Table S16**. Although the incidence
of repeat regions is higher in the full set of NCLV genomes than in Borgs (apparently due to
inclusion of much smaller genomes), the repeats per region and average repeat lengths for Borgs
and NCLVs are closely comparable (revised **Table 2**).

*Overall, the manuscript would be clearer if the **results and discussion were separated**. There
are some nice insights into the Borg protein functions. In the discussion, compare Borgs to **other**
**large elements, plasmids** and viruses.*

We prefer to retain the combined results and discussion, as this would lead to some duplication
of content that could be problematic given the length added in response to reviewer requests.

We have added some additional comparisons to other large elements (phages, plasmids). To our
knowledge, there are no very large archaeal plasmids or archaeal jumbo/ mega viruses reported
to date).

*Although the data is enticing, Borgs have not yet been shown to be viruses. Thus the sentence
in the Abstract “If Borgs are giant archaeal viruses they would fill the gap in the tri(um)virate
of giant viruses of all three domains of life.” is premature, but would be appropriate conjecture in
the discussion following a comparison of Borgs with the largest eukaryotic and bacterial viruses.*

We agree that they are not established as viruses and worded the paper to the best of our ability
to make this clear. Thus, that statement begins with “If”. We feel that this “if” clearly means “If, in
the future, ...”, thus prefer to retain this point.

*Figure 1. Legend Y-axis should read Number of proteins per Borg)*

The caption reads “Number per Borg” and the Caption starts “Number of proteins assigned to
each highly multicopy protein subfamily for each of the 17 Borg genomes...”

**Figure 6. The Nucleocytoviricota included graphs are all in the order Imitervirales. This is**
*just just one of the 3 lineages of giant viruses as noted above.*

Figure 6 features just a few examples. As noted above, we have redone these analyses using a
diverse *Nucleocytoviricota* dataset and updated Table 2.

*Table 2.*

*This again suffers from 64 of the 65 genomes coming from the Imitervirales.*

As above, the analysis has been redone and the Table remade.

Reviewer #4 (Remarks to the Author):

*Authors explored the functional properties of a lineage of very large extrachromosomal and linear*
*double-stranded DNA elements (Borgs) associated with an archaeal genus (Methanoperedens)*
*lacking culture representatives. A main objective was to determine if these intriguing elements,*
*which have a large majority of genes lacking any functional annotation, correspond to mega*
*plasmids or giant viruses. If the later, then it would be the first documented occurrence of such*
*viruses with very large genomes in the archaeal domain. Authors predicted protein 3D structures*
*to expand the functional analysis of multi-copy genes (~10k proteins) among 17 high-quality*
*genomes, described functional families widespread in these genomes, and finally compared*
*genomic features of Borgs and giant eukaryotic viruses. They concluded that “Borgs have*
*numerous features suggestive of lifestyles somewhat analogous to those of giant eukaryotic*
*viruses and we conclude that the weight of evidence now supports a more viral like than plasmid-*
*like lifestyle”.*

*In the first part of results, authors described important functional families widespread in the*
*genomes of Borgs, which provide insights into the functional capability of those elements.*
*Widespread functional families include deSAMPylases and glycoconjugates, which the authors*
*argue might be relevant to a viral lifestyle. Most importantly, authors further explored multi-copy*
*genes annotated as possible viral capsid proteins, already identified by Schoelmerich et al.*
*(2024). The most interesting functional hit is for single-jelly-roll proteins of an archaeal DNA virus*
*named Haloarcula californiae HCIV-1. The spread of genes corresponding to single jellyroll like*
*proteins, their high expression levels, and predicted hexamer assemblies into a capsid-like sheet*
*are certainly interesting and relevant to our understanding of Borgs.*

*In the second part of results, authors compared genomic features of Borgs and giant eukaryotic*
*viruses of the phylum Nucleocytoviricota. They suggest that the common features they identified*
*here and in previous publications (megabase-scale linear genomes with inverted terminal repeats,*
*genomic repertoires for energy metabolism, central carbon compound transformations and*
*translation, and finally pervasive direct repeat regions) correspond to convergent evolution.*

*I have two concerns related to the structure and conclusions of the manuscript, which are*
*described below. Most importantly, authors did not provide convincing evidence (hallmark genes*
*of viruses absent in cellular genomes) that Borgs are viruses.*

*We concur, and note in the manuscript: “..and the lack of a ubiquitous packaging ATPase and for*
*evidence of full capsid assembly limits the argument for classification of Borgs as viruses.”. In*
*fact, the search for such genes motivated the in silico structural biology approach used in the*
*study. Despite in depth investigation, we did not identify any other “hallmark” viral genes. The*
*new analysis involving comparison to BFVD, a large repository of predicted viral protein*
*structures, has been added, but no new insights were obtained. Thus, we do not conclude that*

Borgs are viruses. However, we note that not all viruses have a terminase gene, one of the most
important “hallmark” genes sought in this study. We note that the Borgs PolBs group with
processive PolBs rather than protein primed PolBs. To our knowledge, processive PolBs are not
known to be encoded in MGEs that aren’t viruses (this is now mentioned in the manuscript).

*(1) Lack of signal for hallmark genes of the virion morphogenetic module and limited prominence*
*of single-jelly-roll proteins:*

*Authors did not detect hallmark genes of viruses that are absent in cellular genomes. Critically,*
*single jelly-roll proteins are widespread in cellular genomes*
*(<https://www.pnas.org/doi/full/10.1073/pnas.1621061114>), and only correspond to the major*
*capsid proteins (MCPs) for known viruses with very small genomes.*

In response to this point, we conducted a systematic comparison involving the cellular and
eukaryotic viral SJR proteins in the above-cited paper (Kuprovic and Koonin, Figure 2), Borg SJR
proteins and archaeal viral proteins (identified as top matches by FoldSeek). For almost all cellular
and eukaryotic SJR proteins, the number of aligned atoms (judged based on pruned atom pairs)
was very small, indicating the comparisons are not credible. Only archaeal capsid proteins (6h9c,
6qt9) met the RCSB PDB guidelines, which suggest alignments should include ≥ 40 residues for
reliable comparisons. A summary is provided in **Table S11**. The strongest similarity of Borg
proteins is to capsid proteins from archaeal Sphaerolipoviruses with ~ 30 kbp genomes (not very
small).

Interestingly, the five eukaryotic viral capsid proteins are very distant in structure, possibly due to
the novelty of archaeal virus capsid structures. Some Borg proteins are classified as major capsid-
like, similar to archaeal virus proteins, others align to viral capsid cement proteins. Thus, we
conclude that the identified proteins may be capsid components.

*In viruses with large genomes, the single jelly-roll proteins are cementing the particle (decoration*
*proteins) but do not represent its main component. There are three remaining possibilities: (1)*
*Borgs are the first known viruses with large genomes that only have single jelly-roll proteins as*
*main particle components (single jelly-roll proteins would be the MCPs), (2) their MCPs have not*
*been identified thus far, (3) or they are not viruses (no particle formation). Because single jelly-*
*roll proteins can have cellular functions, no convincing evidence is provided that would*
*demonstrate that Borgs can form particles.*

We agree that these are all interesting possibilities. As there are no established large archaeal
viruses it is difficult to tune expectations. From our understanding, there is precedent for archaeal
viruses with only single SJR proteins as the major capsid components. Specifically, the most
similar protein to Black 801 is 6h9c: Structural basis for assembly of vertical single beta-barrel
viruses. Black 801 aligns well to VP7 major capsid protein (sequence alignment score 71.7,

RMSD over 66 pruned atom pairs is 1.109 Å). This is one of two single jelly-roll major capsid
proteins of *Haloarcula californiae* icosahedral virus 1 (HCIV-1). Further, the predicted pentamer
and hexamer oligomeric states strengthen the case that the Borg SJR proteins are capsid
proteins, as SJR cellular proteins generally do not form these multimers. All top FoldSeek hits are
annotated as viral (capsid) proteins supporting their description as MCP-like.

*(2) Lack of comparisons between Borgs and plasmids*

*Authors provided interesting genomic feature comparisons between Borgs and giant eukaryotic*
*viruses. Yet, no similar comparisons were provided to compare Borgs and plasmids. Since none*
*of the shared features between Borgs and plasmids are described, it is difficult to assess the*
*relevance of a main conclusion of the study: “we conclude that the weight of evidence now*
*supports a more viral like than plasmid-like lifestyle”.*

*Authors may address my concerns by adding perspective on genomic feature similarities and*
*differences between Borgs and plasmids,*

Signatures of plasmids include conjugation machinery, ParAB segregation systems, and certain
types of replication modules. The replication modules of plasmids are diverse but generally do
not resemble cellular replication machinery in the way that Borg and giant virus replication
modules do. An analysis of Borg vs. plasmids was included in the first description of Borgs (Al-
Shayeb et al.), and no new plasmid marker genes came to light in the current study. In general,
there are many features of plasmids and /phages and viruses that are shared, the difference
perhaps resting on the presence or absence of capacity for encapsulation and infection.

There are very few reported plasmids of archaea that we can use for comparison to Borgs.
Motivated by this, we sought and reported plasmids of *Methanoperedens*. The genomes are up
to ~192 kbp (Schoelmerich et al. (2022: *Nature Communications*, 13, 7085
<https://doi.org/10.1038/s41467-022-34588-9>). Given the high relevance of *Methanoperedens*-
specific plasmids for this study, we have added brief information about them. *Methanoperedens*
plasmids predominantly encode proteins involved in nucleic acid manipulation and replication.
Notably, a few plasmid genes involved in translation are not present in the host, pointing to
obligate host dependence on the plasmids. The plasmid genomes tend to occur in relatively
similar copy number to the host, unlike Borgs. The replication modules of plasmids resemble
cellular replication machinery, unlike those of Borg (and giant viruses).

*and by toning down the importance of capsid-like particles detected in Borgs, as those can also*
*be present in cellular genomes for other functions.*

Please see our response above. We have substantially improved our analyses so have not
greatly modulated the language as we still lean towards identification of these proteins as capsid-
like components.

*Of course, these points conflict with the main claim of the study, and the authors may have to*
*adjust the main objectives of the manuscript, which aside from the viral angle certainly provides*
*new insights into the functional capabilities of very intriguing extrachromosomal archaeal DNA*
*elements. Those may or may not be giant viruses, and there are limits to what bioinformatics can*
*deliver. It would be astounding if someone was to observe viral particles with microscopy, thus*
*demonstrating that Borgs are indeed giant archaeal viruses.*

*We agree with these last points and are endeavoring to recover particles, but this is proving to be*
*extremely difficult for entities that comprise a tiny fraction of the biomass in muddy soil.*

Response to reviewer comments round 2:

Reviewer #1 (Remarks to the Author):

I thank the authors for the substantial and constructive revision. I have only a small set of outstanding issues, all focused on data clarity.

-I could not locate the new Supplementary Tables

I am unsure why the Supplementary Tables could not be located as we believe that they were included in the data archive. These tables are now uploaded into the *Nature Communications* site.

Figure 5: The legend lists ipTM values for the capsid pentamers and hexamers. Is this the global ipTM or is the range given for all pair-wise ipTM for all models? Including pair-wise ipTM values is instrumental. Could you include the pairwise values for at least 1 example? Perhaps as a heatmap.

Rather than adding another Supplementary Figure, we have modified the caption to report the range of pairwise ipTM values for subunit-subunit interactions for the best model. The values are reported in the provided .json files.

Moreover, at least the most important PAE plots should be included in the figure to illustrate the modelling quality.

To include the Predicted Aligned Error (PAE) plots would add a very large amount of material to the figure; this information is readily found in the provided .json files. This is now noted in the caption.

Figure S3: As above, specify whether the quoted ipTM is global or interface- specific. Because Zn was incorporated in these models, confirm that exactly one Zn²⁺ per protomer was modelled and note this in the caption. If alternative stoichiometries were explored, indicate which configuration produced the reported scores.

We have modified the caption for supplementary Figure S3 to indicate that the global ipTM is reported. We now note that one Zn²⁺ per protomer was modeled.

In the Supplementary Data file, line 31 reads "ipMT = 0.94"; this should be "ipTM = 0.94".

This has been corrected. Thank you!

Reviewer #2 (Remarks to the Author):

Again, thank you for your input.

Reviewer #3 (Remarks to the Author):

The authors made a number of changes in response to the reviewer comments that improved the manuscript. Most reviewer comments were satisfactorily addressed and the remainder I am willing to go with the author perspective.

Thank you!

Reviewer #4 (Remarks to the Author):

I thank the authors for summarizing some of the main differences between BORGs and plasmids. Since this information was relevant to me, I wonder if the authors could very briefly state in the introduction that major differences are observed between BORGs and plasmids, just before pivoting to discussing features of large eukaryotic viruses. I realize the manuscript is very long, and I only suggest this for the sake of clarity among future readers of the publication.

Thank you for this suggestion. However, this is covered in the later part of the manuscript, so we prefer to not add to (as you note) the already great length.

Overall, I commend the authors for dedicated so many efforts in trying to clarify the nature of those genomic elements using bioinformatics. Despite a lack of signal for some of the most critical viral hallmark genes, the manuscript is full of relevant insights regarding the genomic makeup of BORGs and comparison with evolutionary unrelated viral genomes is both original and relevant.

Finally, I wish the authors best of luck in their efforts to detect particles related to BORGs despite the difficulty of this endeavour.

Tom Delmont

Thank you, Dr. Delmont!